# An improved version of the Piecewise Parabolic Method advection scheme: description and performance assessment in a bidimensional testcase with stiff chemistry in toyCTM v1.0.1.

Sylvain Mailler[1,2], Romain Pennel[1], Laurent Menut[1], and Arineh Cholakian[1]

[1]LMD/IPSL, École Polytechnique, Institut Polytechnique de Paris, ENS, PSL Research University, Sorbonne Université, CNRS, Palaiseau France
[2]École des Ponts-ParisTech, Marne-la-Vallée, France

**Correspondence:** Sylvain Mailler (sylvain.mailler@lmd.ipsl.fr)

**Abstract.** This study presents a novel method to estimate the performance of advection schemes in numerical experiments along with a semi-realistic non-linear, stiff chemical system. This method is based on the examination of the "signature function", an invariant of the advection equation. Apart from exposing this concept in a particular numerical testcase, we show that a new numerical scheme based on a combination of the Piecewise Parabolic Method (PPM) with the flux adjustments of Walcek outperforms both the PPM and the Walcek schemes, for inert tracer advection as well as for advection of chemically active species. From a fundamental point of view, we think that our evaluation method, based on the invariance of the signature function under the effect of advection, offers a new way to evaluate objectively the performance of advection schemes in the presence of active chemistry. More immediately, we show that the new PPM+W ("Piecewise Parabolic Method + Walcek") advection scheme offers chemistry-transport modellers an alternative, high-performance scheme designed for Cartesian-grid Eulerian Chemistry-transport models, with improved performance over the classical PPM scheme. The computational cost of PPM+W is not higher than that of PPM. With improved accuracy and controlled computational cost, this new scheme may find applications in other fields such as ocean models or atmospheric circulation models.

## 1 Introduction

Chemistry-transport models are models that aim at representing the concentration of trace gases and particles in the atmosphere. Many such tools exist, and are used for several purposes, including research and operational forecast. The core of such models consists on a chemical solver adapted to stiff ODE systems along with a framework for solving the advection equation for all the chemical species.

Among the possible strategies to solve the advection equation in chemistry-tranport models are the flux-based advection schemes, based on the ideas of Godunov (1959), including the Van Leer (1977) and Colella and Woodward (1984) schemes. The Walcek (2000) scheme, an improvement of Van Leer (1977), has also been of common use in chemistry-transport models. For example, Geos-CHEM provides an advection framework based on the FV3 module implementing the Putman and Lin (2007) method (Martin et al., 2022), based on the Colella and Woodward (1984) scheme for 1d advection. The Colella and

Woodward (1984) PPM scheme is also implemented in the CMAQ model (Byun and Schere, 2006; Zhao et al., 2020). The CHIMERE model (Menut et al., 2021) also provides the Van Leer (1977) and Colella and Woodward (1984) schemes for horizontal advection, while vertical advection can be treated either with the Van Leer (1977) scheme or the Després and Lagoutière (1999) antidiffusive advection scheme (Lachatre et al., 2020). The Walcek (2000) advection scheme, an improved version of Van Leer (1977) with reduced numerical diffusion is also used in chemistry-transport models including CCATT-BRAMS (Freitas et al., 2012) and LOTOS-EUROS (Timmermans et al., 2022). Therefore, while not the only ones used in chemistry-transport models, the schemes we study here are the base of the numerical resolution of advection in some of the most common chemistry-transport models. Other popular schemes include versions of the Bott (1989) scheme, which is less diffusive than PPM, but has the inconvenient of being non-monotonic, therefore tending to generate extreme values or oscillation in the presence of large concentration gradients (Byun and Schere, 2006), while, by construction, PPM, Van Leer and Walcek enforce mass conservation and monotonicity (Van Leer, 1977; Colella and Woodward, 1984; Walcek, 2000). Despite the availability of higher-order and less diffusive schemes such as the Prather scheme (Prather, 1986) or MPDATA (e.g. Waruszewski et al., 2018), PPM is still considered "highly accurate and efficient enough to be useful" (Harris et al., 2021). A new implementation of the PPM scheme in the CAMx chemistry-transport model has been designed recently to take advantage of the GPU computational facilities (Cao et al., 2023). Since the PPM scheme and, to a lesser extent, the Van Leer (1977) and Walcek (2000) schemes are widely used at least in chemistry-transport modelling, it is important to look for ways to improve these schemes while maintaining their desirable properties of robustness and numerical efficiency.efficiency. In this direction, the goal of the present study is to compare and assess the performance of these schemes in a bidimensional, academic framework including active chemistry, to build an improved version of the PPM scheme, the PPM+W scheme, and to compare the performance of this new scheme to above-cited classical schemes.

In the past, many studies have focused on developing, improving and evaluating advection schemes (e.g. LeVeque (1996), Nair and Lauritzen (2010), Lauritzen et al. (2012), Lauritzen et al. (2014)), but very few studies tackle the evaluation of numerical systems combining advection and chemistry in an academic framework. The most significant step in this direction is the study of Lauritzen et al. (2015), who introduced a toy chemistry scheme mimicking the photolysis and recombination of a virtual stratospheric species. With this simplified non-linear chemical system, they have tested chemistry-advection combinations, and the errors they generate as diagnostics of mass-conservation issues. They have noted that combining such a chemical system with the advection solver may reveal problems that are not generated by inert tracer advection. More recently, Lachatre et al. (2022) have shown that changes in the advection formulation may have significant effects on the behaviour of non-linear chemical processes in the troposphere (in their case, the oxidation pathways of $SO_2$ in a mid-tropospheric volcanic plume). Therefore, as Lauritzen et al. (2015), we feel that it is important to test advection schemes not only with inert tracers but also with active chemistry.

Our goal is to provide such a test case for conditions more representative of tropospheric chemistry at the scale of a urban area, and deploy new tools to evaluate advection schemes in the presence of active chemistry. To meet this objective, apart from classical methods and metrics, we introduce a novel idea, the "signature function", that permits to give a lower bound of model error compared to the exact solution for problems with inert tracer advection, and to isolate the error due to advection itself

in problems including active chemistry. Even though this method is related to the area-coordinate introduced by Nakamura (1996), we introduce a new formulation of this idea along with a way to use it to construct a new error estimate which can be used in problems of pure advection as well as in advection with active chemistry. Apart from this novel way of evaluating advection error in cases with active chemistry, we also propose a new "hybrid" advection scheme, the Piecewise Parabolic Method + Walcek (PPM+W) scheme, made of the PPM scheme with the Walcek (2000) flux adjustments in the vicinity of the extrema.

The PPM+W advection scheme and the concept of signature function are tested within the toyCTM academic chemistry-transport model, already used in Mailler et al. (2021) to test the use of the antidiffusive scheme of Després and Lagoutière (1999) for vertical advection in the atmosphere. The model version used for this study, ToyCTM v1.0 (Mailler and Pennel, 2023), includes horizontal advection with the following schemes (at user's choice): Godunov (1959), Van Leer (1977), Walcek (2000), Colella and Woodward (1984) and PPM+W (present study), while the Després and Lagoutière (1999) scheme is also available for the vertical direction. Chemical processes are solved using a Euler Backward Iterative method (Hertel et al., 1993). As reviewed in Cariolle et al. (2017), this EBI scheme or closely related schemes are used in the MOZART model (Emmons et al., 2010), the ECHAM5-HAMMOZ model (Pozzoli et al., 2008), the TM5 model (Huijnen et al., 2010) and the UKCA climate-composition model (O'Connor et al., 2014; Esentürk et al., 2018).

In Section 2, we describe the flux and chemistry of the numerical experiment we have implemented. In Section 3, we present the set of simulations we have performed and analyzed as well as the description of the advection schemes (3.1), the chemical solver (3.2) and the time-stepping strategy (3.3) implemented in ToyCTM v1.0. In Section 4, we present the concept of signature function that we introduce in this study for the analysis of simulation results. Section 5 compares and discusses the results obtained with the various numerical schemes, and our conclusions are presented in Section 6.

## 2 Numerical experiment description

### 2.1 Chemical mechanism

The chemical mechanism used here (R1-R12) includes a subset of the main reactions of tropospheric gas-phase chemistry. Reactions R1-R3 are the three reactions that constitute the Leighton system, Reactions R4 to R7 account for the formation of the hydroxyl radical OH through the photolysis of ozone in presence of water vapor. Reactions R8 and R9 account for the production of hydroperoxyl radical through oxidation of CO, and the oxidation of NO into $NO_2$ by $HO_2$. Reactions R10-R11 are "termination reactions" that consume the radical species, and R12 describe the final consumption of the $NO_x$ species by formation of nitric acid.

$$NO_2 + h\nu \longrightarrow NO + O \tag{R1}$$

$$O + O_2 + M \longrightarrow O_3 + M \tag{R2}$$

$$NO + O_3 \longrightarrow NO_2 + O_2 \tag{R3}$$

$$O_3 + h\nu \longrightarrow O(^1D) + O_2 \tag{R4}$$

$$O(^1D) + H_2O \longrightarrow OH + OH \tag{R5}$$

$$O(^1D) + N_2 \longrightarrow O + N_2 \tag{R6}$$

$$O(^1D) + O_2 \longrightarrow O + O_2 \tag{R7}$$

$$CO + OH \longrightarrow CO_2 + HO_2 \tag{R8}$$

$$NO + HO_2 \longrightarrow NO_2 + OH \tag{R9}$$

$$HO_2 + HO_2 \longrightarrow H_2O_2 + O_2 \tag{R10}$$

$$OH + HO_2 \longrightarrow H_2O + O_2 \tag{R11}$$

$$NO_2 + OH \longrightarrow HNO_3 \tag{R12}$$

The reaction constants of reactions R1-R12 have been taken mostly from Seinfeld and Pandis (1997), with a temperature of $288\,K$ and pressure of $101325\,Pa$. The photolysis rates rates have been set to typical midday values (e.g. Mailler et al. (2016)):

- $j_{R1} = 8 \times 10^{-3}\,s^{-1}$

- $j_{R4} = 2.5 \times 10^{-5}\,s^{-1}$

In chemistry-transport models, reactions R4-R7 are typically lumped in one single reaction $O_3 + h\nu \longrightarrow OH + OH$ with a pseudo-reaction rate constant that depends on the concentration of air molecules and of water vapor molecules, and applying the quasi steady-state approximation to $O(^1D)$. For this study, we have chosen to treat $O(^1D)$ as a prognostic species to preserve the full chemical stiffness of the problem, with lifetimes ranging from $\simeq 4 \times 10^{-9}\,s$ for $O(^1D)$ to several days for CO.

Of course key processes like oxidation of methane and of other volatile organic compounds are not taken into account in the above mechanism, but it retains some key features of troposphric chemistry, which we think important:

- Extreme stiffness

- OH production depends on the presence of ozone, water vapor and sunlight

- Non-linear behaviour of ozone production (in this simplified system, ozone production depends on the simultaneous presence of nitrogen oxides, OH, and available CO for oxidation).

## 2.2 Definition of Test Case

### 2.2.1 Simulation domain

The simulations are performed on a domain $\mathcal{D} = [0; L] \times [0; L] \times [0; H]$, where $L = 10^5\,\mathrm{m}$ and $H = 1000\,\mathrm{m}$. Since we will use only barotropic winds, the problem is in fact bidimensional in $x$-$y$, with no $z$ dependance. However, the choice has been made to treat formally the problem as tridimensional in order to be able to use quantities such as air density and reaction rate constants with their usual magnitudes and units. Due to the barotropic nature of the problem, discretization in the vertical direction is in one single cell, while the $x$ and $y$ dimensions are split evenly into $n = 25$ subintervals each. This corresponds to a resolution $\delta x = 4 \times 10^3\,\mathrm{m}$, rather typical for regional-scale chemistry-transport modelling. Domain $\mathcal{D}$ is therefore discretized into $n^2$ cells, each cell with thickness $H$ and horizontal section $\delta x^2 = \frac{L^2}{n^2}$.

### 2.2.2 Wind field

The flow we use in this study is the swirling deformational flow introduced by LeVeque (1996) (their Eqs. 9.5-9.6):

$$u = \frac{L}{T}\sin^2\left(\frac{\pi x}{L}\right)\sin\left(2\pi y\right)g\left(t\right); \; v = -\frac{L}{T}\sin^2\left(\frac{\pi y}{L}\right)\sin\left(2\pi x\right)g\left(t\right), \tag{1}$$

with

$$g(t) = \cos\left(\frac{\pi t}{T}\right). \tag{2}$$

$T$ is the half-period of the experiment, and the design of the flow is such as all fluid particules are back at their original location after time $T$, but in-between thay have undergone a deformation, which is maximal at time $\frac{T}{2}$. Here, while the LeVeque (1996) study formulates the problem with non-dimensional scales for time and space, we set a dimensional scale-length $L = 10^5\,\mathrm{m}$ and half-period $T = 86\,400\,\mathrm{s}$. The velocity field corresponding to these values is depicted on Fig. 1. Equation 1 ensures that the wind is zero at domain boundaries ($x \in \{0, L\}$ or $y \in \{0, L\}$) so that no mass enters nor leaves domain $\mathcal{D}$. Therefore, no boundary conditions for concentrations are needed.

The time-dependant streamfunction for this flow is:

$$\psi\left(x, y, t\right) = -\frac{L^2}{\pi T}\sin^2\left(\frac{\pi x}{L}\right)\sin^2\left(\frac{\pi y}{L}\right)g\left(t\right) \tag{3}$$

## 2.3 Initial conditions

The numerical experiments will be conducted in the domain $\mathcal{D}$ defined above, with the chemical scheme described above. To define the initial conditions, we introduce a concentration profile (between 0 and 1) as follows:

$$\varphi\left(x; y\right) = \sin^2\frac{2\pi x}{L}\sin^2\frac{2\pi y}{L} \text{ if } x < \frac{L}{2} \text{ and } x < \frac{L}{2} \tag{4}$$

$$\varphi\left(x; y\right) = 0 \text{ otherwise.} \tag{5}$$

The initial conditions are defined as follows (in terms of mixing ratio):

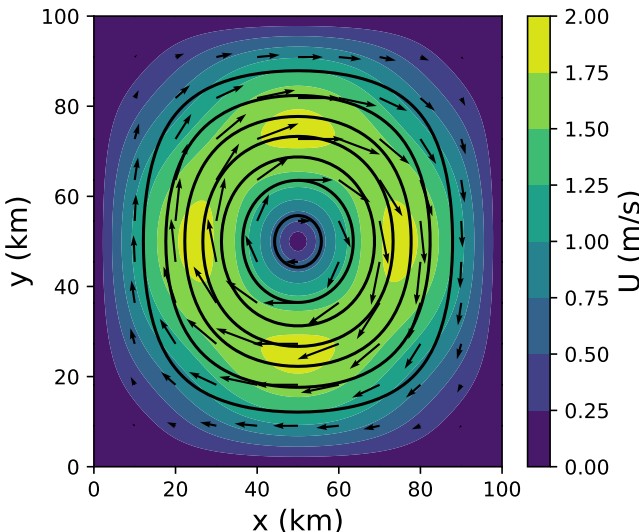

**Figure 1.** streamlines (black contours), wind vectors and wind module in $\mathrm{m\,s^{-1}}$ (color shades) at $t = 0$ for the swirling deformational wind field defined in Eq. 1 with $L = 10^5\,\mathrm{m}$ and $T = 86,400\,\mathrm{s}$

– $\alpha_{\mathrm{TRC}} = 100\,\mathrm{ppb} \times \varphi(x; y)$ (see Fig. 2)

     – $\alpha_{\mathrm{O3}} = 30\,\mathrm{ppb}$

     – $\alpha_{\mathrm{CO}} = 500\,\mathrm{ppb}$

     – $\alpha_{\mathrm{NO}} = 100\,\mathrm{ppb} \times \varphi(x; y)$

     – $\alpha_{\mathrm{NO2}} = 10\,\mathrm{ppb} \times \varphi(x; y)$

– $\alpha_{\mathrm{H2O}} = 8.044 \times 10^{-3}$

For water vapor, mixing ratio $\alpha_{\mathrm{H2O}} = 8.044 \times 10^{-3}$ corresponds to a specific humidity of $5\,\mathrm{g\,kg^{-1}}$. The initial concentrations of the other active species (O, O $\left(^1\mathrm{D}\right)$, OH, HO$_2$, CO$_2$, H$_2$O$_2$, HNO$_3$) are initialized to zero. They will be produced by reactions R1, R4, R5, R8 and R12.

Finally, another species of inert tracer, TRCb, is introduced so that:

$\alpha_{\mathrm{TRCb}} + \alpha_{\mathrm{NO}} + \alpha_{\mathrm{NO2}} = 110\,\mathrm{ppb}.$                                              (6)

This species is designed so that, at the initial time and later along the run, in the exact solution (but not necessarily in numerical solutions), all along the run, the sum $\alpha_{\mathrm{TRCb}} + \alpha_{\mathrm{NO}} + \alpha_{\mathrm{NO2}} + \alpha_{\mathrm{HNO3}}$ is uniform, constant and equal to $110\,\mathrm{ppb}$.

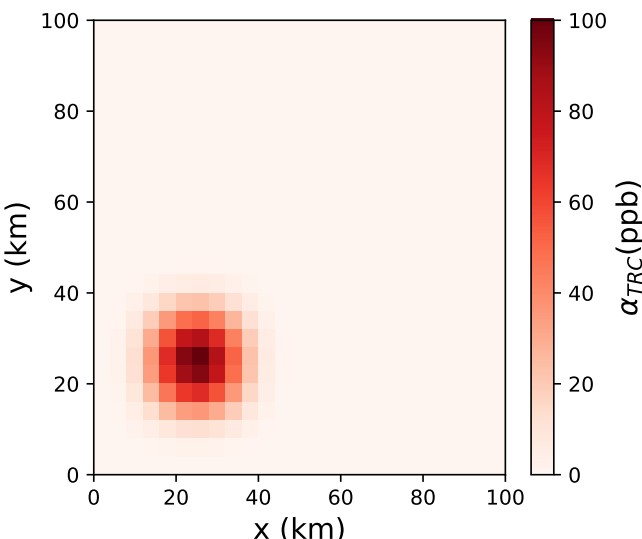

**Figure 2.** Initial mixing ratio of TRC, proportional to $\varphi(x;y)$ (defined in 5), discretized on domain $\mathcal{D}$ with n=25 subintervals ($\delta x = \delta y = 4\,000\,\text{m}$). Note that, at $t = 0$, $\alpha_{\text{NO}} = \alpha_{\text{TRC}}$ and $\alpha_{\text{NO2}} = \frac{\alpha_{\text{TRC}}}{10}$ have the same spatial distribution.

## 3 Numerical methods

### 3.1 Advection schemes

#### 3.1.1 Existing advection schemes

The following existing advection schemes have beeen tested in the study:

1. Godunov (1959)

2. Van Leer (1977)

3. Walcek (2000)

4. PPM (Colella and Woodward, 1984)

These schemes are flux-based, upwind-biased, semi-Lagrangian schemes based on polynomial reconstructions of the average concentrations. These polynomial reconstructions are piecewise-constant for Godunov (1959), piecewise linear for Van Leer (1977) and Walcek (2000) (Fig. 3a,b), and piecewise parabolic except in the vicinity fo the extrema for for PPM (Fig. 3c). The Van Leer (1977) scheme exhibits a discontinuity in the vicinity of the maximum, with the concentrations having a positive jump towards the maximum (Fig. 3a). As a consequence of this discontinuity, due to the upwind-biased strategy, the fluxes going out of the maximum (from the high side of the discontinuity) will be systematically overestimated compared to the fluxes going into the maximum (from the low side of the discontinuity), thereby tending to advect too much mass out of the maximum,

and not enough mass into the maximum. Walcek (2000) presents his scheme as a way to counteract this bias by adjusting the flux estimates in the cells next to the maximum, in order to intentionnally overestimate the fluxes going into the maximum to counteract the excessive estimation of the fluxes out of the maxima (Fig. 3b).

### 3.1.2 The PPM+W scheme

The PPM scheme presents the same caveat as Van Leer in the vicinity of extrema, with a strong discontinuity on each side of the extremum (Fig. 4a), with the effect of underestimating the mass flux into the maximum, and to overestimate the mass flux out of the maximum. Therefore, since Walcek (2000) has proven that his flux adjustments in the vicinity of the extrema is successful in improving the Van Leer scheme, it makes sense to try applying the same flux adjustments to PPM, which seems to have a behaviour similar to Van Leer (1977) in the vicinity of mixing ratio extrema (Fig. 3a,c). To test this idea, we design a new scheme based on PPM, but applying the Walcek flux adjustments in the vicinity of the extrema (Figs. 3d and 4b). We call this scheme PPM+W, standing for "Piecewise Parabolic Method + Walcek flux adjustments". The PPM+W has the same behaviour as Walcek in the extrema and the neighbouring cells, and the same behaviour as PPM in all other cells.

We detail here the procedure applied for this scheme. Let $\ldots \alpha_{i-2}, \alpha_{i-1}, \alpha_i, \alpha_{i+1}, \alpha_{i+2}, \ldots$ be the values of mixing ratio in the model cells numbered by the 1d index $i$, with a Courant number $\nu$. For the sake of simplicity we assume that $\delta x = 1$. The objective of this procedure is to calculate the average mixing ratio between $x_{i+\frac{1}{2}}$ and $x_{i+\frac{1}{2}} - \nu$, $\widetilde{\alpha}_{i+\frac{1}{2}}$ (the rest of the implementation of advection from this estimate is detailed in Lachatre et al. (2020)).

The procedure is as follows.

- If $(\alpha_{i+1} - \alpha_i)(\alpha_{i+2} - \alpha_{i+1}) > 0$ and $(\alpha_{i-1} - \alpha_{i-2})(\alpha_i - \alpha_{i-1}) > 0$:
   the current cell is not a neighbour of a maximum. We estimate $\widetilde{\alpha}_{i+\frac{1}{2}}$ following the Piecewise Parabolic Method procedure described in Colella and Woodward (1984).

- Otherwise: we estimate the Walcek-adjusted flux as:
   $s = \text{sign}(\alpha_{i+1} - \alpha_i) \text{Min} \left( \frac{1}{2} |\alpha_{i+1} - \alpha_{i-1}|, 2|\alpha_{i+1} - \alpha_i|, 2|\alpha_i - \alpha_{i-1}| \right)$, the Van Leer slope
   if $(\alpha_{i+1} - \alpha_i)(\alpha_{i+2} - \alpha_{i+1}) <= 0$: $\beta = 1.75 - 0.45\nu$
   else: $\beta = \text{Max}(1.5, 1.2 + 0.6\nu)$
   $\widetilde{\alpha}_{i+\frac{1}{2}} = \alpha_i + \frac{1}{2}(1 - \nu) \times \beta \times s$.

The $\beta$ coefficient ($\beta > 1$ by construction) is introduced by Walcek (2000) to steepen the Van Leer slopes in the vicinity of the maxima to obtain the desired overestimation of tracer fluxes into the maxima. These steepened slopes are visible on Fig. 3b for the Walcek (2000) scheme and on Figs. 3d and 4b for the PPM+W scheme.

### 3.1.3 Computational cost of the advection schemes

To evaluate the computational cost of these advection schemes, advection of a 1d vector composed of $2 \times 10^5$ cells has been performed over 520 time steps, corresponding to $1.04 \times 10^8$ calls to the reconstruction routine, plus the update of the mixing-ratio values at each time step. The calculation time for all these advection schemes is presented in Table 1, showing that the

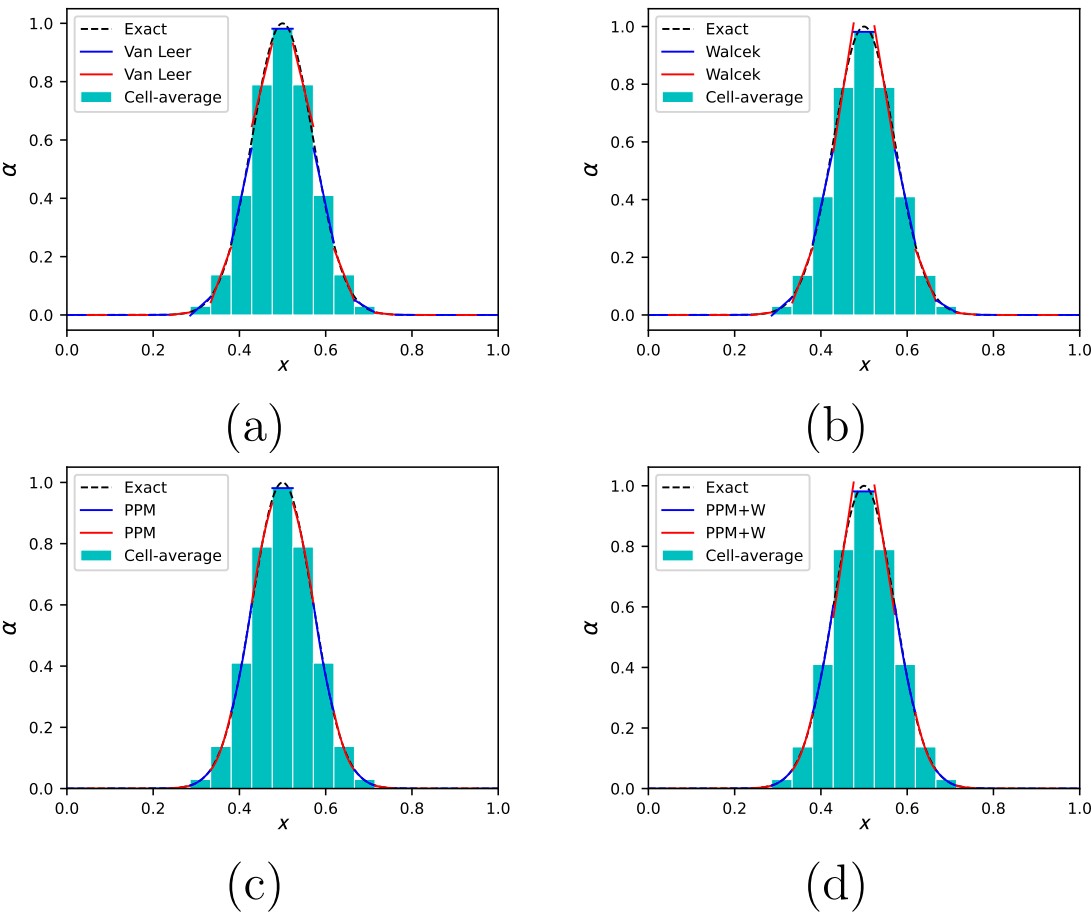

**Figure 3.** Reconstruction of a Gaussian mixing ratio profile by (a) the Van Leer (1977) scheme; (b) the Walcek (2000) scheme; (c) the PPM scheme and (d) the PPM+W scheme. The $x$-axis is a non-dimensional space coordinate. The reconstruction has been performed for a Courant number $\nu = 0.4$. The reconstructed fields are presented with alternating red-blue colors to enhance the discontinuities between neighbouring cells.

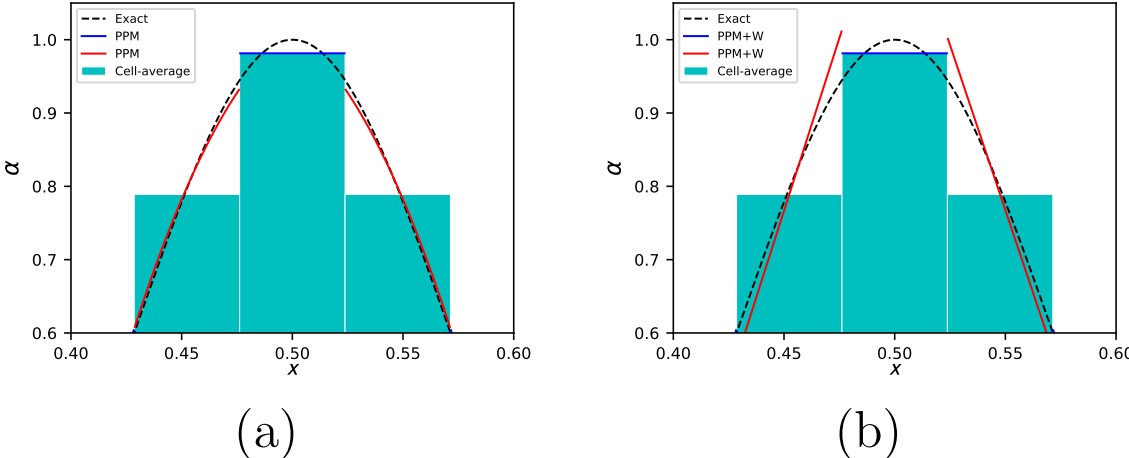

**Figure 4.** (a) Same as Fig. 3c but zoomed in the vicinity of the mixing ratio maximum, and (b) same as Fig. 3d but zoomed in the vicinity of the mixing ratio maximum.

| Scheme | Execution time |
|--------|---------------|
| Godunov | 5.8 ns |
| Van Leer | 12.2 ns |
| Walcek | 14.9 ns |
| PPM+W | 30.3 ns |
| PPM | 32.4 ns |

**Table 1.** Mean calculation time per cell and per timestep for the five advection schemes retained for the present study. The calculation has been performed in Fortran, a programming language frequently used for operational chemistry-transport models, on a laptop with an Intel Core i7-1165G7 CPU.

schemes using linear reconstruction (Van Leer and Walcek) are less costly than the schemes using parabolic reconstruction (PPM and PPM+W), due to the simpler calculation. Interestingly, the computation cost of PPM+W is slightly smaller than the cost of PPM, possibly because in the cells neighbouring a concentration extremum the reconstruction is linear in PPM+W instead of parabolic in PPM (Fig. 4).

### 3.1.4 Convergence properties

A first comparison between the PPM+W advection scheme and the other four tested schemes has been performed in terms of numerical convergence. For this purpose, an inert tracerwith a squared cosine bell distribution (initially) has been advected over a 1d periodic domain of unit length, with a constant and uniform speed, with a unit duration. This convergence test has been performed dividing the domain into 10, 20, 40, 80, 160 and 320 grid cells, and with a Courant number 0.5. The results of this convergence test are shown on Figure. 5. Several features can be observed on Figs. 5. The convergence rates, defined as

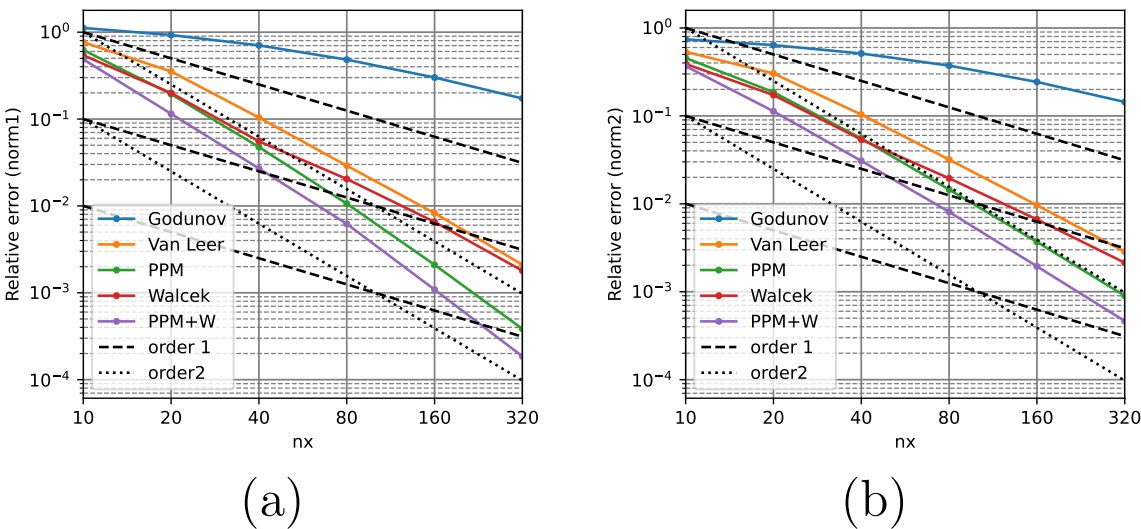

**Figure 5.** (a) $\|\cdot\|_1$-error as a function of the number of $n_x$ for the five advection schemes used in the present study; and (b) same as (a) for $\|\cdot\|_2$-error

| scheme | $\|\cdot\|_1$ | $\|\cdot\|_2$ |
|---|---|---|
| Godunov | 0.80 | 0.76 |
| Van Leer | 1.97 | 1.76 |
| Walcek | 1.86 | 1.64 |
| PPM | 2.45 | 2.03 |
| PPM+W | 2.55 | 2.07 |

**Table 2.** convergence rate of the five advection schemes used in the present study for $\|\cdot\|_1$ and $\|\cdot\|_2$

the opposite of the log-log slope between the two last data points ($n_x = 160$ and $n_x = 320$) are given in Table 2. From these results, several observations can be made.

    First, the PPM+W advection scheme performs better on this simple convergence test than the other tested schemes. In particular, throughout all the resolution range, the error obtained both in $\|\cdot\|_1$ (Fig. 5a) and $\|\cdot\|_2$ (Fig. 5b) with the PPM+W scheme is 30 to 50% lower than with the classical PPM scheme. This difference persists even for high resolutions. On the

contrary, the Walcek scheme strongly outperforms the Van Leer scheme for coarse resolutions, but this difference tends to diminish for higher resolution. In other terms, it looks like the Walcek flux corrections when applied to the Van Leer scheme permit to improve accuracy when model resolution is coarse, but that the same flux corrections when applied to the PPM scheme improves accuracy even for fine resolutions. Finally, the convergence rate for PPM+W is similar to that of PPM, in our case around 2.5 for $\|\cdot\|_1$ convergence rate and around 2 for $\|\cdot\|_2$ (Fig. 2).

## 3.2 Chemistry solver: the Euler Backward Iterative method

The stiff chemical system is integrated using an Euler Backward Iterative method (EBI). As described in Hertel et al. (1993); Cariolle et al. (2017), we obtain the concentration vector $\mathbf{c}\left(t + \delta t_{chem}\right)$ as the solution of:

$$\mathbf{c}\left(t + \delta t_{chem}\right) = \frac{\mathbf{c}\left(t\right) + \delta t_{chem}\mathbf{P}\left(t + \delta t_{chem}\right)}{1 + \delta t_{chem}\mathbf{L}\left(t + \delta t_{chem}\right)}. \tag{7}$$

For the present study, the focus is to test the performance of the advection scheme in articulation with active chemistry. Due to this focus, we limit errors in the resolution of the chemical system by using a short time step for chemistry ($\delta t_{chem} = 20\,\mathrm{s}$).

Equation 7 is a non-linear, fixed point equation, and can be solved only numerically, usually with an iterative method. Formally, Eq. 7 guarantees exact mass conservation. However, this is true only if a very good convergence of the solution is reached (Cariolle et al., 2017). To limit violation of mass-conservation in our study, we have set a very strict convergence criterion for the iterative resolution of Eq. 7, stopping iteration when the estimate of $\mathbf{c}\left(t + \delta t_{chem}\right)$ yields a relative difference less than $\varepsilon$ for each species between $\mathbf{c}\left(t + \delta t_{chem}\right)$ and $\frac{\mathbf{c}(t) + \delta t_{chem}\mathbf{P}(t + \delta t_{chem})}{1 + \delta t_{chem}\mathbf{L}(t + \delta t_{chem})}$. The convergence parameter is set to $\varepsilon = 10^{-4}$ in the UKCA chemistry-transport model (Esentürk et al., 2018), and $\varepsilon = 10^{-6}$ in the present study. This very strict convergence criterion is in line with the short chemical time step to obtain the best possible numerical solution of the chemical evolution of the system, even to the cost of slow computations. Again, this choice is due to the purpose of this study to test the performance of the advection scheme, limiting as much as possible the errors in the chemical solver.

## 3.3 Time-stepping

The advection time step $\delta t_{adv}$ has been set to $\delta t_{adv} = 1800\,\mathrm{s}$. With maximal wind module $U \simeq 1.8\,\mathrm{m\,s^{-1}}$ (Fig. 1) and $\delta x = 4 \times 10^3\,\mathrm{m}$, this yields a maximal Courant number $\nu_{\max} \simeq 0.8$. The time-stepping strategy follows a Strang-style time stepping (Strang, 1968), with the steps as follows:

1. Integrate chemistry over $\frac{\delta t_{adv}}{2}$ (45 chemical time steps)

2. Integrate zonal advection over $\frac{\delta t_{adv}}{2}$

3. Integrate meridional advection over $\delta t_{adv}$

4. Integrate zonal advection over $\frac{\delta t_{adv}}{2}$

5. Integrate chemistry over $\frac{\delta t_{adv}}{2}$ (45 chemical time steps)

Table 3 summarizes the 6 simulations that have been performed. The above-described case with chemical reactions R1-R12, initial conditions as described in 2.3, domain shape and discretization as described in 2.2.1, advection schemes as described in 3.1 and chemical solver as described in 3.2. Along with the simulations performed with each of the 5 advection schemes, a "Base" simulation has been performed with the same setup as the other simulations but without advection. As discussed later (Section 4), this Base simulation will serve as a benchmark to estimate advection errors in the other five simulations.

| Label | Chemistry | Mass flux | Advection scheme | $\delta t_{chem}$ | $\delta t_{adv}$ | Duration |
|--------|-----------|-----------|------------------|-------|-------|----------|
| Base | (R1-R12) | – | – | 20 s | – | 86400 s |
| Godunov | (R1-R12) | Eq. 3, $L = 10^5$ m, T=86400 s | Godunov (1959) | 20 s | 1800 s | 86400 s |
| Van Leer | (R1-R12) | Eq. 3, $L = 10^5$ m, T=86400 s | Van Leer (1977) | 20 s | 1800 s | 86400 s |
| Walcek | (R1-R12) | Eq. 3, $L = 10^5$ m, T=86400 s | Walcek (2000) | 20 s | 1800 s | 86400 s |
| PPM | (R1-R12) | Eq. 3, $L = 10^5$ m, T=86400 s | Colella and Woodward (1984) | 20 s | 1800 s | 86400 s |
| PPM+W | (R1-R12) | Eq. 3, $L = 10^5$ m, T=86400 s | PPM+W | 20 s | 1800 s | 86400 s |

**Table 3.** Summary of the main characteristics of the simulations that have been performed.

### 3.4  Conservation properties

It is worth noting that, by construction, flux-based advection integration is mass-conservative since the mass flux out of a cell through a facet is compensated exactly by the mass flux into the neighbouring cell through the same facet. Equation 7 also guarantees mass conservation in the Euler Backward scheme as soon as the chemical reactions themselves are balanced (Hertel et al., 1993) (which is the case of reactions R1-R12 except for the imbalance of dioxygen in R8, due to integrating reaction $H + O_2 \longrightarrow HO_2$ into the kinetically limiting time step $CO + OH \longrightarrow CO_2 + H$).

However, due to the finite number of iterations in the iterative resolution of Eq. 7, mass-conservation is only enforced with a finite precision $\varepsilon = 10^{-6}$ (see Section 3.2). Therefore, the relative mass imbalance in the outputs for C, active N (without taking into account $N_2$) and H can be expected to be of the order of $\varepsilon$, since the Euler-Backward Iterative scheme (Eq. 7) is in principle mass-conservative, but mass conservation is obtained "only if a good convergence of the solution is reached" (Cariolle et al., 2017).

Mass calculations for C, active N, H and TRC have been performed between the beginning and the end of the simulations. The result of this calculation for the PPM+W and Base simulations are given in Table 4, showing that the relative mass imbalance at the end of simulation is $\simeq 10^{-6}$ for active N, $\simeq 10^{-12}$ for C and H, and $\simeq 10^{-15}$ for TRC (which has no chemistry and is therefore affected only by advection which, as discussed above, ensure mass conservation up to numerical accuracy for the flux-based schemes we have implemented).

The imbalance results are similar for all simulations, including the Base simulation which has no advection, which shows that the small mass imbalance for chemically active species (up to $\simeq 10^{-6}$) is essentially due to the finite precision in the Euler Backward Iterative chemistry solver.

In summary, the integration strategy we introduce above permits to conserve mass (up to numerical accuracy for inert species and up to an arbitrary numerical tolerance defined by the user, in our case $\varepsilon = 10^{-6}$, for chemically active species), and

conserves initially uniform mixing ratios up to numerical accuracy for the species with no active chemistry. All the advection schemes presented above are also built to respect monotonicity: they do not create new mixing-ratio extrema.

| Species | Composition | Relative imbalance (PPM+W) | Relative imbalance (Base) |
|---------|-------------|----------------------------|---------------------------|
| C | $CO+CO_2$ | $1.1\times10^{-12}$ | $7.4\times10^{-13}$ |
| Active N | $NO+NO_2+HNO_3$ | $4.0\times10^{-7}$ | $2.5\times10^{-7}$ |
| H | $2H_2O + OH + HO_2 + 2H_2O_2 + HNO_3$ | $7.4\times10^{-13}$ | $6.5\times10^{-13}$ |
| TRC | TRC | $8.9\times10^{-16}$ | $0$ |

**Table 4.** Mass-conservation diagnostic for C, active N, H and TRC in the PPM+W and Base simulations.

## 4 The signature function

### 4.1 Accuracy of inert tracer concentrations

As the LeVeque (1996) flow is designed so that at $t = T$ every Lagrangian particle is back to its original location, it is possible
to estimate the accuracy of numerical simulation by comparing the final simulated tracer concentration field to its initial value, therefore giving access to the magnitude of numerical error.

In the present study, we will estimate model error in $\|\cdot\|_1$, introducing $E_1$ as the normalized $\|\cdot\|_1$ error on mixing ratio:

$$E_1 = \frac{\sum_{i=1}^{N} \rho_i V_i \left| \alpha_i^t - \alpha_i^0 \right|}{\sum_{i=1}^{N} \rho_i V_i \alpha_i^0}, \tag{8}$$

where index $i$ spans the entire domain. In the present case where density $\rho_i$ and cell volume do not vary acrcoss cells, Eq. 8
boils down to:

$$E_1 = \frac{\sum_{i=1}^{N} \left| \alpha_i^t - \alpha_i^0 \right|}{\sum_{i=1}^{N} \alpha_i^0}, \tag{9}$$

### 4.2 The signature function for inert tracer advection

We introduce here a new idea to evaluate advection schemes. As far as we know, this idea has not been tested in the past literature, but resembles the area-coordinate formulation used by Nakamura (1996).
Let us imagine a fluid with density $\rho(x; y; z; t)$ in a three-dimensional domain $\mathcal{D}$, advecting a tracer having initially a mixing ratio $\alpha(x; y; z; t = 0) \in [0; \infty[$, following equations:

$$\frac{\partial \rho}{\partial t} + \nabla(\rho \mathbf{u}) = 0 \tag{10}$$

$$\frac{\partial \alpha}{\partial t} + \mathbf{u} \cdot \nabla \alpha = 0. \tag{11}$$

For any given time $t$ and any mixing ratio $0 \leq X \leq 1$, we can define $\mathcal{S}^t(X)$ as the mass of fluid in the volume $\mathcal{D}^t(X)$ defined as the set of all $(x; y; z)$ where tracer mixing ratio $\alpha(x; y; z; t) < X$, divided by the entire mass of fluid in $\mathcal{D}$:

$$\mathcal{S}^t(X) = \frac{\int_{\mathcal{D}} H(X - \alpha(x; y; z; t)) \rho d\mathcal{V}}{\int_{\mathcal{D}} \rho d\mathcal{V}}, \tag{12}$$

where H is the Heaviside step function ($H(u) = 1$ if $u > 0$; $H(u) = 0$ if $u \leq 0$). The $\mathcal{S}^t$ function can be, in some sense, interpreted as a mass-weighted cumulative probability density function of tracer mixing ratio. If we reduce this definition to 2d flows with uniform density, $\mathcal{S}^t$ is related to the reciprocal function of the area-coordinate formulation of Nakamura (1996).

With this definition, we always have $S^t(0) = 0$ (for all $t$), and $S^t(X) \to 1$ when $X \to +\infty$ (more precisely, $S^t(X) = 1$ as soon as $X$ is larger than the maximum value of $\alpha(x; y; z; t = 0)$ over domain $\mathcal{D}$.

Eq. 12 makes clear that function $\mathcal{S}^t$ is invariant during the motion: For any given value of X, $\mathcal{S}^0(X)$ is the (normalized) mass of the fluid parcel $\mathcal{D}^0(X)$ that has a tracer mixing ratio $\alpha < X$ at $t = 0$. We can observe that, since mixing ratio in Lagrangian parcels is preserved by pure advection (Eq. 11), $\mathcal{D}^t(X)$ and $\mathcal{D}^0(X)$ represent the same Lagrangian fluid parcel at a different time. We also know that the mass of fluid in Lagrangian parcels is constant in time due to mass conservation for the carrier fluid (Eq. 10). Since the total mass of fluid in $\mathcal{D}$ is also constant in time for the same reason, this implies that, for all $t$ and $X$, $\mathcal{S}^t(X) = \mathcal{S}^0(X)$. In other words, the signature function $\mathcal{S}^t$ is a time-invariant of the advection equation. The invariance of this function holds for both divergent and non-divergent flows.

Therefore, since we know that for the exact solution $\mathcal{S}^t = \mathcal{S}^0$, the departure of the numerical evaluation of function $\mathcal{S}^t$ from $\mathcal{S}^0$ in numerical simulations can be used as an objective measurement of discretization errors.

In practice, in a Eulerian model discretized in $N$ cells, each cell has an evaluation of tracer mixing ratio $\alpha_i^t$. The evaluation of $\mathcal{S}^t$ is straightforward:

$$\mathcal{S}^t(X) \simeq \frac{\sum_{i=1}^{N} H(X - \alpha_i) \rho_i V_i}{\sum_{i=1}^{N} \rho_i V_i}. \tag{13}$$

Generally, the numerical evaluation of $\mathcal{S}^t$ in a Eulerian model will differ from $\mathcal{S}^0$: the signature of the initial tracer distribution will evolve under the effect of the errors of the advection scheme, and the magnitude of the signature modification can serve as a measure of the extent of advection error. In the particular case in which the flow is non-divergent and the carrier fluid mass $\rho_i V_i$ is the same in all model cells, norm-1 difference between $\mathcal{S}^t$ and $\mathcal{S}^0$ can be calculated as:

$$\int_0^\infty |\mathcal{S}^t(X) - \mathcal{S}^0(X)| \, dX = \frac{1}{N} \sum_{i=1}^{N} \left| \widetilde{\alpha^t}_i - \widetilde{\alpha^0}_i \right|, \tag{14}$$

where $\widetilde{\alpha^t}_i$ is the vector of all mixing ratios in model cells *sorted in increasing order*. This simplified formula holds only in the specific case in which the flow is invariant and the mass of fluid is distributed evenly between all the model cells. In this case, and only in this case, it is also convenient to introduce the normalized norm-1 difference $\mathcal{E}_1$ between $\mathcal{S}^t$ and $\mathcal{S}^0$ as:

$$\mathcal{E}_1 = \frac{\sum_{i=1}^{N} \left| \widetilde{\alpha^t}_i - \widetilde{\alpha^0}_i \right|}{\sum_{i=1}^{N} \widetilde{\alpha^0}_i}. \tag{15}$$

Figure 6 shows an example of how comparing the $\mathcal{S}^t$ signature function with $\mathcal{S}^0$ permits to compare the accuracy of two simulations at a time when no analytic solution is easily accessible. Panels 6a-6b show the mixing ratio for TRC at $\frac{T}{2}$ in the Godunov and Van Leer simulations, respectively. Without access to the exact solution, it is hard to compare quantitatively the quality of these simulations at that stage, even though indicators such as the maximal tracer mixing ratio can give a partial

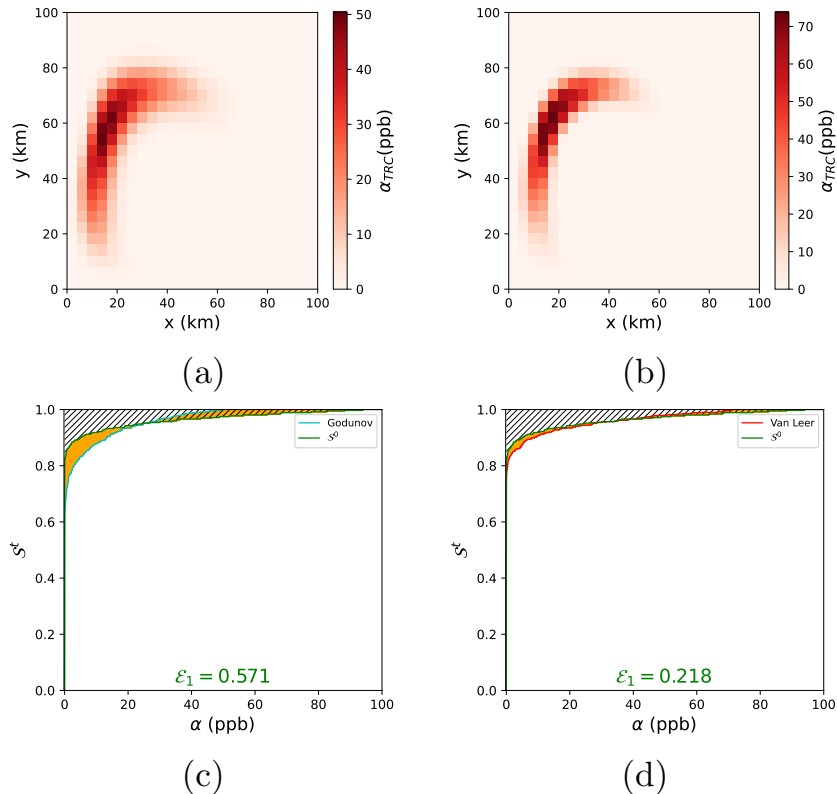

**Figure 6.** (a) TRC mixing ratio as simulated in simulation Godunov at time $\frac{T}{2}$; (b) same as (a) for the Van Leer simulation. (c) Signature function for TRC mixing ratio in the Godunov simulation compared to the Base simulation ($\mathcal{S}^0$, green line); and (d) same as (c) for the Van Leer simulation. Signature error $\mathcal{E}_1$ is equal to ratio of the shaded area between the representative curves of $\mathcal{S}^0$ and $\mathcal{S}^t$ to the hatched area left of $\mathcal{S}^0$.

information. Panels 6c-6d show the $\mathcal{S}^{\frac{T}{2}}$ function compared with $\mathcal{S}^0$ for the Godunov and Van Leer simulations, respectively. This graphical representation permits to give an intuitive meaning to the normalized signature error $\mathcal{E}_\infty$ as the total area between

the representative curves of $\mathcal{S}^t$ and $\mathcal{S}^0$ (shaded in Figs. 6c-d), divided by the total area left of the representative curve of $\mathcal{S}^0$ (hatched in Figs. 6c-d). This measure gives an indication of model error based not only on one particular point representative of a part of the tracer distribution (*e.g.* the maximal value, a very partial indicator), but an integrated error indicator taking into account the maximal and minimal values as well as the entire tracer distribution in-between these values. In the case presented in Fig. 6c-d, the area between $\mathcal{S}^0$ and $\mathcal{S}^t$ is smaller in the Van Leer simulation (Fig. 6c, with $\mathcal{E}_\infty = 0.218$) than in the Godunov

simulation (Fig. 6d, with $\mathcal{E}_\infty = 0.571$).

### 4.3 The signature function for advection of active species

The concentrations of active species evolve not only under the effect of advection but also due to chemical reactions. Therefore, the time-invariance of the signature function does not hold for these species. However, the signature function can still be used to compare and evaluate simulations if one remarks that, in the case we study here with no variations in air density and air temperature, and with no emissions, the chemistry that takes place in each Lagrangian air parcel is independant of its position. Therefore, for all species, the signature function at time $t$, $\mathcal{S}^t$, should theorectically be the same as in the Base simulation with no advection. If we note $\bar{\mathcal{S}}^t$ the signature function at time $t$ in the simulation without advection, for all $t$ we have $\mathcal{S}^t = \bar{\mathcal{S}}^t$: At any time, signature of the distribution of every chemical species should be the same in the case with advection as in the case without. In other words, advection should only deform the map of all chemical species, but not change the chemistry within each Lagrangian air parcel. With a non-linear chemical system as Reactions R1-R12, there is no easy access to the exact solution of the system even without advection: $\bar{\mathcal{S}}^t$ is not known exactly for the chemically active species. However, simulation Base resolves the chemical reactions with exactly the same chemical solver as the other simulations, but without motion. Therefore, comparing the distribution of a chemical species at time $t$ in a simulation with advection to its distribution in the Base simulation without advection will permit to have an estimate of how the numerical errors on advection affect the distribution of chemically active species.

As an illustration of this, Fig. 7 shows the ozone mixing ratio at $\frac{T}{2}$, as simulated in three simulations with advection (7a-7c), and the Base simulation without advection (7d). These figures alone do not make it easy to discriminate between the three numerical simulations. Fig. 8 shows the signature function for the same three simulations (Van Leer, PPM and PPM+W): this time, both the visualization of the agreement between $\mathcal{S}^t$ and $\bar{\mathcal{S}}^t$ and the objective calculation of $\mathcal{E}_1$ clearly shows that the PPM+W simulation agrees better with the Base simulation in terms of signature function ($\mathcal{E}_1 = 0.0622$), followed by the PPM simulation ($\mathcal{E}_1 = 0.0679$) and the Van Leer simulation ($\mathcal{E}_1 = 0.0787$). Apart from this quantitative agreement, some qualitative and local conclusions can be drawn from the graphical comparisons between $\mathcal{S}^t$ and $\bar{\mathcal{S}}^t$, such as the fact that the representation of the ozone minimum in the PPM+W simulation is more accurate than in the PPM simulation, which is visible in the initial part of the signature function ($\alpha_{O3} < 30\,\mathrm{ppb}$) where the shaded area is smaller in PPM+W (Fig. 7c) than in PPM and Van Leer.

## 5 Results and discussion

### 5.1 Results on classical performance indicators

Fig. 9 shows the evolution of some performance indicators along the experiments. For the same reasons as for the signature function, these metrics should be the same in all simulations in the absence of numerical errors in the representation of advection, so that the differences between the time series obtained in the simulations with advection and the Base simulation reveal the effects of numerical errors in the representation of advection. It is interesting to note that all the metrics represented in Fig. 9 can be derived directly from the signature function.

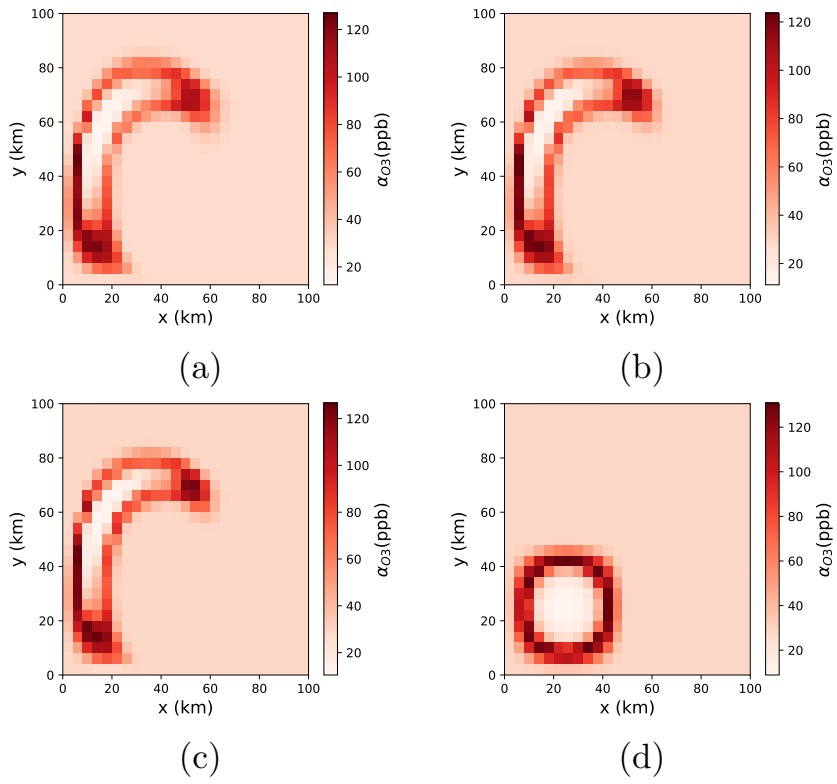

**Figure 7.** Ozone mixing ratio at $\frac{T}{2}$ as simulated in (a) the Van Leer simulation; (b) the PPM simulation, (c) the PPM+W simulation and (d) the Base simulation.

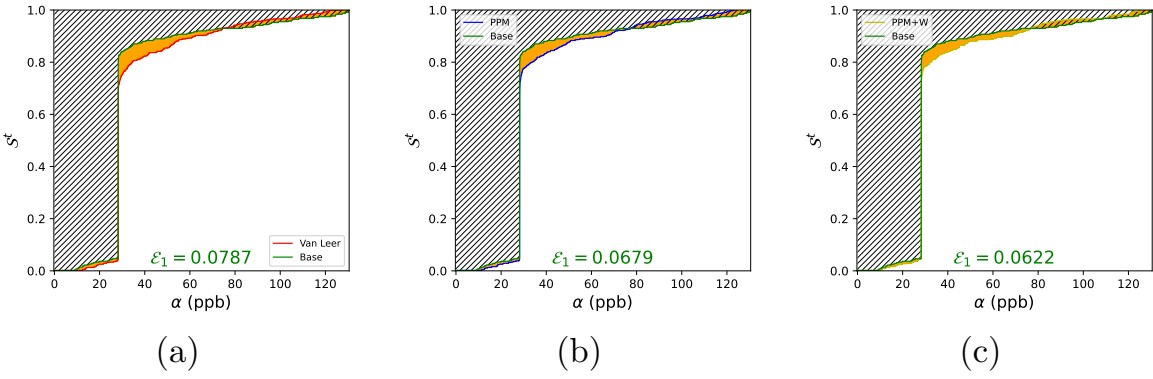

**Figure 8.** Signature function for the $O_3$ mixing ratio at $\frac{T}{2}$, $\mathcal{S}^{\frac{T}{2}}$, as simulated in (a) the Van Leer simulation; (b) the PPM simulation and (c) the PPM+W simulation. As is Fig. 6, signature error $\mathcal{E}_1$ is equal to ratio of the shaded area between the representative curves of $\mathcal{S}^0$ and $\mathcal{S}^t$ to the hatched area left of $\mathcal{S}^0$

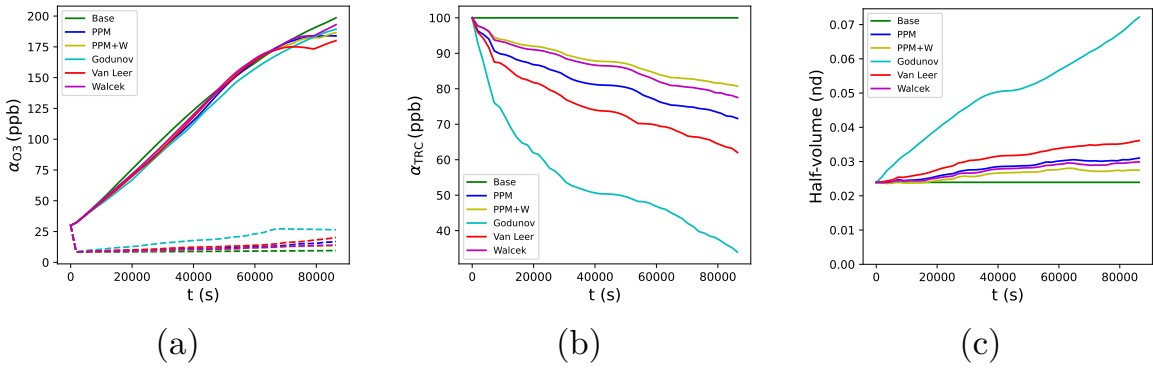

**Figure 9.** Time series for all the simulations for: (a) Minimum and maximum of the $O_3$ mixing ratio, (b) maximum TRC mixing ratio and (c) half-volume of the TRC plume (relative to the volume of the entire domain)

Regarding ozone extrema along the simulation (Fig. 9a), we observe smaller differences between the different simulations. The PPM+W, PPM and Walcek simulations produce very comparable values of ozone maxima. Surprisingly, the simple Godunov schemes represents slightly better the ozone maximum than all other schemes except Walcek towards the end of the simulation. However, the representation of the ozone minimum by the Godunov simulation is very bad, failing to represent a minimum lower than the background value towards the end of the simulation. The PPM+W and Walcek simulations represent the smallest (and closest to the Base simulation) minimum value for $O_3$, followed by PPM and Van Leer.

Regarding the preservation of tracer maxima (Fig. 9b), the PPM+W simulations performs best, with a clear edge over the Walcek and PPM simulations (Fig. 9b). For this metric, the Walcek simulation gives better results than the PPM simulation, with a smaller half-plume, closer to the Base simulationWalcek and PPM. The PPM and Walcek simulations perform similarly for this metric.

Figure 9c shows evolution of the non-dimensional half-volume of the tracer plume during the simulation (defined as the smellest volume containing half of the mass of TRC, divided by the total volume of domain $\mathcal{D}$). As discussed in Lachatre et al. (2020), this is a measure of the diffusivity of the advection schemes. With this metric, we see that the PPM+W simulation performs best, followed by Walcek and PPM.

### 5.2 Accuracy and signature error

Unlike the partial metrics presented in Fig. 9, the normalized $\|\cdot\|_1$ signature error of mixing ratios $\mathcal{E}_1$ is a performance diagnostic for the simulations that takes into account the entire distribution of mixing ratios, and not just particular values such as the maximum or minimum. Fig. 10 shows $\mathcal{E}_1$ for TRC, $O_3$, NO and $NO_2$ for all the simulations, the smallest $\mathcal{E}_1$ indicating the best performing simulation.

The first information we get from Fig. 10 is that, for all these compounds, the PPM+W simulation performs best in this regard. For the case of ozone (Fig. 10d), the $\mathcal{E}_1$ time series discriminates much more between the different simulations, with a clear superiority of the PPM+W simulation over the Walcek and PPM simulations, while the differences between these three

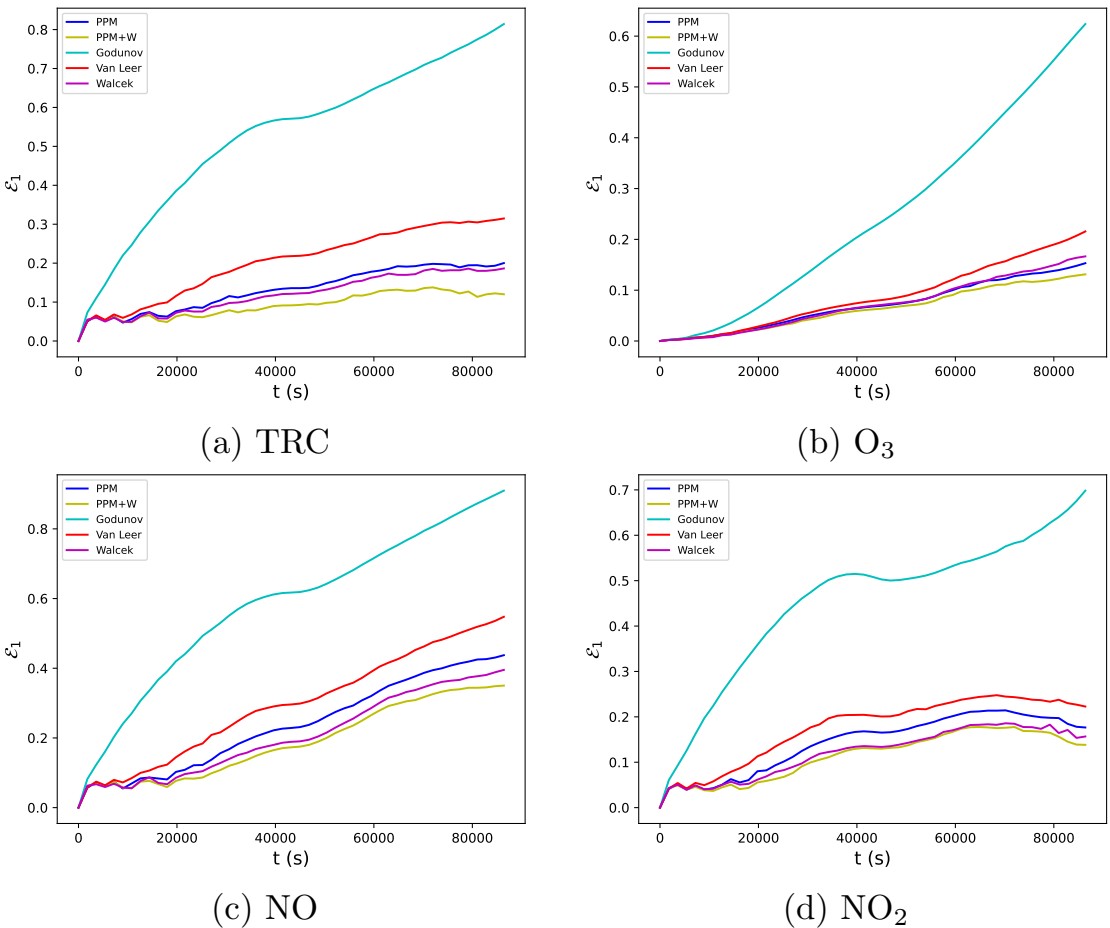

**Figure 10.** Time series for $\mathcal{E}_1$ ($\|\cdot\|_1$ normalized signature error) for (a) TRC; (b) $O_3$; (c) NO and (d) $NO_2$

simulations in the ozone min-max plot (Fig. 9b) appeared small. Analyzing the evolution of $\mathcal{E}_1$ throughout the simulation shows
that, for TRC, NO and $NO_2$ the best performing simulations are PPM+W, Walcek and PPM, in this order, but that for ozone
PPM performs better than Walcek, a conclusion that could not be drawn from the min-max plot (Fig. 9b), which indicated a
better performance of Walcek regarding both the ozone maximum and the ozone minimum.

While we have shown so far that the analysis of $\mathcal{E}_1$ permits to draw clearer conclusions regarding the performance of the
various simulations, in the present case we can confirm these results by comparing $\mathcal{E}_1$ to a more classical metric, normalized
$\|\cdot\|_1$ error $E_1$ (Eq. 9) of the simulations with advection relative to the Base simulation without advection. Unlike $\mathcal{E}_1$, $E_1$ can
only calculated at the final time step (when all the Lagrangian particles are back to their initial location), since the exact solution
for $t < T$ is not accessible. Several observations can be made from the values of $E_1$ and $\mathcal{E}_1$ in Table 5.

First, in all cases we always have $\mathcal{E}_1 < E_1$. Qualitatively, this can be interpreted as $\mathcal{E}_1$ being a weakened form of $\|\cdot\|_1$ error,
retaining the differences in the distribution of mixing ratios, but eliminating the differences in the location of the tracer plume.

|  | Normalized $\|\cdot\|_1$ error | | | | $\mathcal{E}_1$ | | | |
|---|---|---|---|---|---|---|---|---|
|  | TRC | O3 | NO2 | NO | TRC | O3 | NO2 | NO |
| Godunov | 0.864 | 0.668 | 0.781 | 0.927 | 0.814 | 0.624 | 0.698 | 0.910 |
| Van Leer | 0.408 | 0.262 | 0.419 | 0.594 | 0.315 | 0.216 | 0.223 | 0.548 |
| Walcek | 0.243 | 0.200 | 0.312 | 0.425 | 0.186 | 0.167 | 0.157 | 0.395 |
| PPM | 0.291 | 0.193 | 0.337 | 0.472 | 0.200 | 0.153 | 0.177 | 0.437 |
| PPM+W | **0.207** | **0.164** | **0.283** | **0.372** | **0.120** | **0.131** | **0.138** | **0.350** |

**Table 5.** Normalized $\|\cdot\|_1$ error $E_1$ (Eq. 9) and normalized $\|\cdot\|_1$ signature error $\mathcal{E}_1$ (Eq. 15) at the end of the simulations for $O_3$, NO, $NO_2$ and TRC, compared to the Base simulation with no advection. In each column, the lowest error value is in bold font, the second-lowest in underlined.

Interestingly, the performance ranking between the five simulations obtained by analyzing the signature error $\mathcal{E}_1$ is the same for all variables as with $E_1$: for TRC, NO, $NO_2$, PPM+W performs best, followed in this order by Walcek, PPM, Van Leer and Godunov, but for $O_3$, PPM performs better than Walcek. In our study case, analyzing $\mathcal{E}_1$ permits to compare the performance of the various simulations without access to the exact solution, and giving the same results as the analysis of $E_1$ (which requires access to the exact solution). Even for simulations with comparable performance as the Walcek and PPM simulations, the signature error $\mathcal{E}_1$ permits to diagnose which of the two simulation performs better for each variable. Even though the differences between these two simulations are not drastic, and depend on the interest species, the conclusions drawn from the analysis of signature error $\mathcal{E}_1$ are the same and with $E_1$. This gives confidence in the usability of signature error $\mathcal{E}_1$ as a proxy of simulation accuracy when the exact solution is not available.

Having verified this, it is useful to go back to Fig. 10 and interpret the evolution of $\mathcal{E}_1$ in time as a hint of when error appears along the simulation. In this regard, we see two very different behaviours between the analyzed variables. For TRC, NO and $NO_2$, substantial errors appear almost instantly after $1800\,\mathrm{s}$ of simulation (one single time step). This is due to the action of the wind field on the initially very sharp peak of these species (Fig. 2). On the contrary, for $O_3$, having an initially uniform distribution, errors due to advection appear only when sufficient heterogeneity is introduced in the $O_3$ map by chemical processes, since all the advection schemes are built to advect exactly a uniform mixing ratio, maintaining its uniformity. Therefore, the onset of advection errors on $O_3$ is much slower than on the three species that have initially heterogeneous distributions.

### 5.3 Results at 1 km-resolution

To test the impact of higher resolution on our results, we have performed the same test case as above but refining the resolution from $4\,\mathrm{km}$ to $1\,\mathrm{km}$, accordingly reducing the time step from $1800\,\mathrm{s}$ to $450\,\mathrm{s}$. The results for this higher resolution simulation are shown in Table. 6. This table shows that, also at this resolution, the PPM+W offers the best performance of all the tested schemes, but, unlike in the $4\,\mathrm{km}$ resolution case, this does not hold for all species: at 1 km resolution, the PPM scheme performs comparably to PPM+W for inert tracer TRC, and slightly better for $NO_2$. However, PPM+W performs clearly better for $O_3$

| | Normalized $\|\cdot\|_1$ error | | | | $\mathcal{E}_1$ | | | |
|---|---|---|---|---|---|---|---|---|
| | TRC | O3 | NO2 | NO | TRC | O3 | NO2 | NO |
| Godunov | 0.434 | 0.275 | 0.357 | 0.584 | 0.386 | 0.244 | 0.237 | 0.564 |
| Van Leer | 0.0417 | 0.0486 | 0.0521 | 0.0905 | 0.0241 | 0.0418 | 0.0325 | 0.0773 |
| Walcek | 0.0202 | 0.0303 | 0.0423 | 0.0509 | 0.016 | 0.0244 | 0.0344 | 0.0472 |
| PPM | 0.0169 | 0.0336 | **0.0324** | 0.0574 | **0.00881** | 0.0221 | **0.0263** | 0.0382 |
| PPM+W | **0.0148** | **0.0247** | 0.0383 | **0.0483** | 0.00901 | **0.0112** | 0.031 | **0.0332** |

**Table 6.** Normalized $\|\cdot\|_1$ error $E_1$ and normalized $\|\cdot\|_1$ signature error $\mathcal{E}_1$ **in the 1 km simulations** at the end of the simulations for O$_3$, NO, NO$_2$ and TRC, compared to the Base simulation **at 1 km resolution** with no advection. In each column, the lowest error value is in bold font, the second-lowest in underlined.

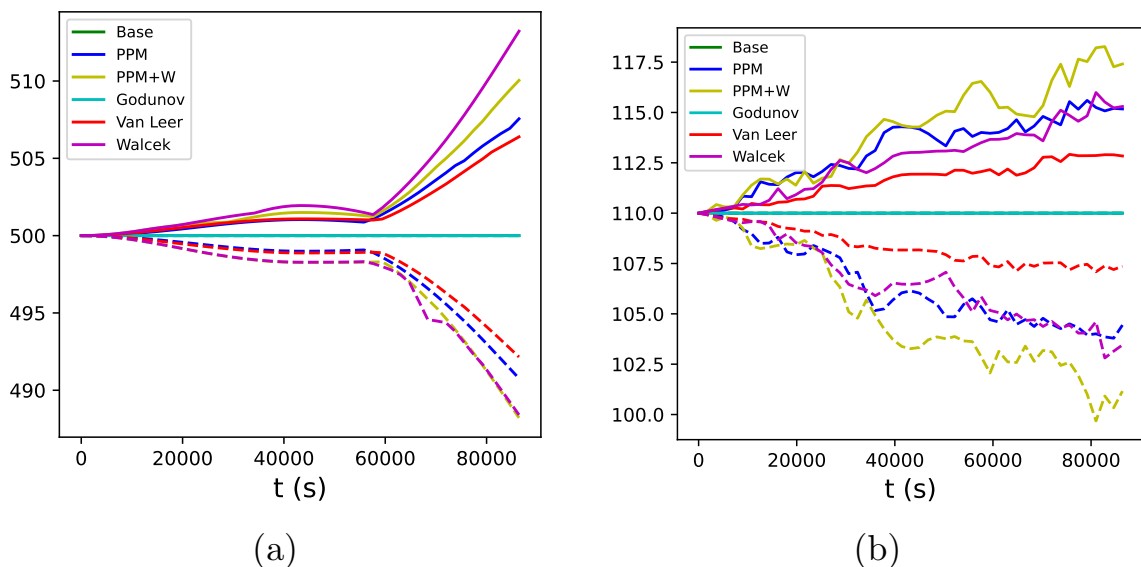

**Figure 11.** Minimum and maximum values in the 4 km simulations (presented in the manuscript) for: $\alpha_{CO} + \alpha_{CO_2}$ (panel (a)) and $\alpha_{NO} + \alpha_{NO_2} + \alpha_{HNO_3} + \alpha_{TRCb}$ (panel (b))

and NO. These results suggest that the improvement brought by PPM+W over PPM may tend to become smaller for higher resolutions (contrary to what we observe in Fig. 5).

## 5.4 Conservation of sum species

As shown by Godunov (1959), "among [linear] schemes of second order accuracy [for the advection equation], there is none which satisfies the monotonicity condition". As a consequence of this result, known as the Godunov theorem, second-order, monotonic schemes for the advection equation are all non-linear. In particular, among all the schemes we study here, all are

non-linear due to their specific treatment of the maxima to preserve monotonicity. However, these non-linear effects should be kept as small as possible. To test this property, we have tested the conservation of two sum species, which should stay uniform and constant along the run:

- $\alpha_{\mathrm{TRCb}} + \alpha_{\mathrm{NO}} + \alpha_{\mathrm{NO_2}} + \alpha_{\mathrm{HNO_3}}$, where TRCb is an inert tracer whose initial distribution is defined by Eq. 6. The mixing ratio of this sum species should stay constant, uniform and equal to $110\,\mathrm{ppb}$.

- $\alpha_{\mathrm{CO}} + \alpha_{\mathrm{CO_2}}$, which should stay constant, uniform and equal to $500\,\mathrm{ppb}$ due to the conservation of carbon.

Figure 11 shows that, as imposed by the Godunov theorem, only the Godunov (1959) scheme preserves exactly the uniformity of sum species. Other schemes are higher-order and monotonic, but therefore non-linear, and do not preserve the uniformity of these sum species. The magnitude of the departure of the mixing ratio for these species from its theoretical value is a measure of the non-linearity of advection schemes, which is an undesirable property since the advection equation itself is linear. Figure 11 shows that non-linearities up to $1-2\%$ appear for all schemes in the representation of $CO + CO_2$, and that they tend to be strongest in the Walcek and the PPM+W schemes. Regarding $\mathrm{TRCb} + \mathrm{NO} + \mathrm{NO_2} + \mathrm{HNO_3}$, the non-linearities reach up to $10\%$ of the expected value, and are strongest in the PPM+W scheme. These results suggest that non-conservation of sum species in the vicinity of mixing ratio extrema, which is observed in all second-order monotonic schemes, tends to be stronger when the Walcek (2000) flux corrections are applied, which might represent a drawback of PPM+W compared to PPM. In spite (or due to) this increasingly non-linear behaviour in the vicinity of the extrema, PPM+W preserves the values of extrema better than PPM (Fig. 9b).

## 6 Conclusions

We have introduced the signature function $\mathcal{S}^t$ as a sort of mass-weighted cumulative probability distribution function of tracer mixing ratio (Eq. 12) and shown that $\mathcal{S}^t$ is an invariant of the advection equation. This invariant is not a scalar as other classical invariants (tracer mass, minimum and maximum mixing ratio etc.) but a function, therefore containing much more information than the above-cited, more classical invariants. In fact, these invariants (tracer mass, minimum and maximum mixing ratios etc.) can be derived directly from $\mathcal{S}^t$ so that $\mathcal{S}^t$ can be considered as a "stronger" invariant. As other invariants such as the minimum and maximum values of mixing ratio, it is usually not preserved perfectly by the advection schemes, and the degree of non-conservation of $\mathcal{S}^t$ gives a proxy of model error. Based on this idea, we propose the normalized $\|\cdot\|_1$ signature error $\mathcal{E}_1$ (Eq. 15) as a measure of model error. Graphically, $\mathcal{E}_1$ is the normalized area between the simulated curve of $\mathcal{S}^t$ and its theoretical curve (e.g. Fig. 6c-d).

In this context, we have shown that the signature function and its normalized $\|\cdot\|_1$ error $\mathcal{E}_1$ can also be used as an error estimate for chemically active species, even though $\mathcal{S}^t$ is not an invariant of the system in this case. This is achieved by comparing a simulation including chemistry and advection to a "companion" simulation with chemistry only but no advection (the Base simulation in the present study). For each chemical species and at any time $t$, these two simulations should in theory

have the same $\mathcal{S}^t$, but in practice they don't, due to errors in advection. Therefore, errors due to advection can be estimated even for chemically active species by calculating the $\mathcal{E}_1$ error between the two functions (Fig. 8).

We have used this new invariant in order to evaluate a new advection scheme that we have designed for this study, based on the PPM scheme (Colella and Woodward, 1984) with flux-corrections in the vicinity of the extrema as in Walcek (2000) (Figs. 3-4). This new advection scheme, which we propose to name PPM+W (for Piecewise Parabolic Method + Walcek),

has been tested for both an inert tracer and chemically active species, with a velocity flux from LeVeque (1996) (Eq. 1). A simplified chemical scheme with 12 reactions has been designed, representing daytime photochemistry of nitrogen oxides in the presence of CO, including short-lived species such as $O(^1D)$ or O (Reactions R1-R12). With an initial peak of NO and $NO_2$ concentrated over a small area (Fig. 2) and initially uniform concentrations of CO and $O_3$, this case generates a sharp ozone minimum colocalized with the $NO_x$ peak, a large area with background $O_3$ concentration, and a belt of high ozone

concentrations in-between (Fig. 7). With this test case, we have evaluated the PPM+W scheme along with the PPM scheme, the Walcek (2000) scheme, the Van Leer (1977) scheme and the Godunov (1959) scheme. In this case, we have shown that, for all species and all the metrics we have tested, the PPM+W scheme performs better than all the other tested schemes (Table 5), with a normalized $\|\cdot\|_1$ error $E_1$ of 16.4% on $O_3$ (19.3% for PPM), 20.7% on TRC (24.3% with Walcek), 28.3% for $NO_2$ (31.2% for Walcek) and 37.2% for NO (42.5% for Walcek). Table 5 also shows the $\mathcal{E}_1$ signature error, which is always

smaller than $E_1$. Interestingly, examining the $\mathcal{E}_1$ errors for the same variables and the same simulations yields exactly the same conclusions as examining normalized $\|\cdot\|_1$ error $E_1$. This shows that, even without access to an exact solution, the $\mathcal{E}_1$ signature error permits to compare the simulations against each other for inert and reactive species, giving the same conclusions as an accuracy analysis with $E_1$. This being shown, Fig. 10 permits to visualize the evolution of error along the simulation, while $E_1$ can be calculated only at $t = T$, because the LeVeque (1996) flux field is designed to guarantee that at that time all the

Lagrangian particles should be back to their initial locations.

Thefore, the conclusion of this study is twofold. First, regarding the signature function as an invariant of the advection equation, we feel that this invariant contains much more information than other invariants that have been typically used to check advection schemes, such as the minimum or maximum values of mixing ratios, while not requiring access to the exact solution. In the case of chemistry-transport models, generalizing this concept to more dimensions (by studying the mass-weighted prob-

ability distribution function of all species simultaneously rather than one signature function per species) could be promising. For the same reasons as exposed above, this multidimensional probability distribution function should be an invariant of the advection equation. This concept could be explored to quantify model error in a synthetic way across all variables, instead of separately for each variable. The approch introduced here with the $\mathcal{E}_1$ signature error could possibly be generalized by using statistical tools such as the Wasserstein distance to compare these multidimensional probability distribution functions with each

other. The main limit to the use of the signature function as a tool to evaluate the advection framework in real-world geophysical models is that it relies on mixing ratio conservation, which holds for pure advection but does not hold in the presence of mixing or diffusion. However, some geophysical compartments like deep ocean or the stratosphere are substantially affected by mixing only for very long time scales, so that the signature function could be a useful tool to verify the behaviour of advection frameworks in such compartments.

From a more applied point of view, the PPM+W advection scheme introduced here performs better than both the Walcek (2000) scheme and the PPM scheme, with all the metrics we have tested and for both inert and active species. It not only preserves the tracer maxima better than the Walcek (2000) scheme (Fig: 9), but is also more accurate than the PPM scheme for the representation of ozone and other active species (Table 5). Of course, these results are proven in the present study only on one 2d test case with active chemistry and on a 1d convergence test. While the results of both the test-case and the convergence test consistently indicate a better performance of PPM+W compared to the other advection schemes tested here, they do not include the full range of Courant numbers and tracer patterns that occur in realistic models. For example, an additional numerical experiment presented here (Section 5.3 and Table 6) shows that, when the resolution is refined four times compared to our main test case, the advantage of PPM+W over PPM seems to be reduced, and for one species ($NO_2$), PPM+W is even outperformed by PPM in terms of accuracy. However, even in this high-resolution test case, the performance of PPM+W is better than that of PPM for the three other tested species. The convergence test performed in this study (Fig. 5) suggests that there is no systematic reduction of the performance of PPM+W at higher resolution. A possible drawback of the PPM+W scheme when compared to the PPM scheme is its stronger non-linearity in the vicinity of maxima (Fig. 11), which permits a better conservation of the maxima themselves, but induces more numerical artifacts in the representation of sum species in the vicinity of extrema.

We have also observed (Table 1) that the computation cost of PPM+W is slighty lower than the cost of the PPM scheme, which is used in some of the most popular chemistry-transport models. The improved performance in terms of accuracy and of preservation of tracer extrema without increasing the computational load makes this scheme a very interesting option for chemistry-transport models, in an effort to reduce numerical diffusion, which is important in particular in the presence of non-linear chemistry as discussed in Lachatre et al. (2022)).

*Data availability.* No dataset has been used for this study.

*Code availability.* This study has been performed using ToyCTM v1.0.1 (Mailler and Pennel, 2023). All the Python scripts used to launch the model and to perform the post-processing of model outputs to obtain the figures in this paper and the numbers in Table 5 are available from https://doi.org/10.5281/zenodo.10018761.

The Fortran code AdvBench v1.0.0 used to evaluate advection performance (Table 1) is available from https://doi.org/10.5281/zenodo.7937121.

ToyCTM v1.0.1, AdvBench v1.0.0 and all the scripts used to lanch the model and post-process its outputs for the present study are distributed under the GNU General Public License v2.0.

*Author contributions.* All the authors have contributed to the design of the simulated cases; SM has performed and analyzed the simulations, SM has developed the software with RP.

*Competing interests.* None.

*Acknowledgements.* This study has been supported by ADEME (Agence de l'Environnement et de la Maitrise de l'Énergie) under grant ESCALAIR.

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
