# Peer review of "An improved version of the Piecewise Parabolic Method advection scheme: description and performance assessment in a bidimensional testcase with stiff chemistry in toyCTM v1.0.1."

_Geoscientific Model Development, 2023_

## Author Comment (AC1)

Answer to the Reviewer Comment signed by Christopher Walcek 10.5194/gmd-2023-78-RC1 on the manuscript "**An improved version of the Piecewise Parabolic Method advection scheme: description and performance assessment in a bidimensional testcase with stiff chemistry in toyCTM v1.0.**"

S. Mailler, R. Pennel, L. Menut, A. Cholakian

July 28, 2023.

We are grateful to Christopher Walcek for his careful reading and insightful comments on our manuscript. Two main comments by Pr. Walcek regard the need to discuss how a higher-resolution would affect our results, and the question of the "inability to preserve lumped but conserved species". We address both these comments in detail below.

**Contents**

**1 Comments by C. Walcek**

The authors introduce a modification and improvement of numerical advection algorithms by merging existing techniques in a way the optimizes the numerical approximation of advection. They also introduce a new error metric that assesses not only local extrema, but also the distribution and range of concentration distributions that are advected. My main complaint about the study is the limited nature of the test presented (only one Courant number, and one fairly well-resolved gaussian hill shape. Aside from a few minor pounts that need to be mentioned, the paper is publishable.

**1.** typo line 250. How can fluid density everywhere be zero. is this a typo ?

**2.** using these deformational flows, figures like 8c and others do not properly quantify the true "accuracy" of the scheme since the shape is advected/sheared into a shape that has scale features that are smaller than the 4 km resolution of this experiment. If this entire study were performed at 1 or 0.1 km resolution, the error measures would change. Basically, the simulation called "base" in these figures is not a true average of the 4x4km resolution "base". When the peak of the "base" Guassuan shape is sheared into successively smaller and smaller cells, then averaged BACK to the 4 km coarse T=0 4km-resolution discrete grid, then the simulation called "base" should change with time also. This applies to the figures which show the distribution at the T/2 times at the point where the. Here the authors could do a simulation at spatial resolution of 0.4 km (much smaller than 4 km), then MAP the 0.4 km simulation back onto the 4km "base" grid by averaging 10X10 grid cells into a single 4 km grid, then compare.

**3.** one of the problems with the Walcek peak scheme is the inability to preserve lumped but conserved species since there it is guaranteed that numerical treatment around local maxima are NOT treated identical

| | Normalized $\|\cdot\|_1$ error | | | | $\mathcal{E}_1$ | | | |
|---|---|---|---|---|---|---|---|---|
| | TRC | O3 | NO2 | NO | TRC | O3 | NO2 | NO |
| Godunov | 0.434 | 0.275 | 0.357 | 0.584 | 0.386 | 0.244 | 0.237 | 0.564 |
| Van Leer | 0.0417 | 0.0486 | 0.0521 | 0.0905 | 0.0241 | 0.0418 | 0.0325 | 0.0773 |
| Walcek | 0.0202 | 0.0303 | 0.0423 | 0.0509 | 0.016 | 0.0244 | 0.0344 | 0.0472 |
| PPM | 0.0169 | 0.0336 | **0.0324** | 0.0574 | **0.00881** | 0.0221 | **0.0263** | 0.0382 |
| PPM+W | **0.0148** | **0.0247** | 0.0383 | **0.0483** | 0.00901 | **0.0112** | 0.031 | **0.0332** |

Table 1: Normalized $\|\cdot\|_1$ error $E_1$ and normalized $\|\cdot\|_1$ signature error $\mathcal{E}_1$ **in the 1 km simulations** at the end of the simulations for $O_3$, $NO$, $NO_2$ and TRC, compared to the Base simulation with no advection. In each column, the lowest error value is in bold font, the second-lowest in underlined.

in SUMS of conserved species. The "peaks" in NO, $NO_2$ and $HNO_3$ will occur at different places than the combined $NO+NO_2+HNO_3$. In the context of the problem presented here, total nitrogen should be conserved ($NO+NO2+HNO3$). The authors should be able to show that there is non-monotonic behavior of any advection scheme which treats local EXTREMES algebraically differently. Please show graphs of $NO+NO_2+HNO_3$.

**2 General answer**

Pr. Walcek judges this paper publishable, with one main complaint about "the limited nature of the test presented (only one Courant number, and one fairly well-resolved gaussian hill shape)".

We agree that the fact that there is only one test presented could be a limit, however the manuscript presents objective scores for two different metrics and 4 different species on a total of 5 simulations, amounting to 40 metric scores, and these different species test the advection schemes in different conditions because their distributions are very different. We also introduce one new advection scheme (that has to be explained) and a different metric with the idea, as Pr. Walcek describes it, to assess "not only local extrema, but also the distribution and range of concentration distributions that are advected". This has guided the choice to perform only one test-case (but with several advection configurations and chemical species).

It is true that there is "only one Courant number", which does evolve in time though since the mass-flux (and therefore the CFL) is equal to zero at $\frac{T}{2}$. For the same reasons as above, we did not perform a sensitivity test on timestep.

**These limits/questions will be acknowledged in the manuscript for a revised version.**

Regarding the fact that our test case consists essentially of a "fairly well-resolved gaussian hill shape", this is essentially true for the inert tracers, even at half-time under the influence of wind shear (Fig. 5 of the manuscript), but this is not true for the ozone concentrations. Fig. 6 of the manuscript (reproduced here as Fig. 1) shows that the high-ozone belt that circles the plume takes an extremely elongated shape under the effect of wind shear and of the different chemical regimes. Fig. 1 clearly shows that this high-ozone belt has a croos-dimension of the order of magnitude of one single grid-cell, which is extremely challenging for the advection framework.

**3 Specific answers**

On point **1.**, equation $\rho(x;y;z;t=0)$ is not properly written, as the Reviewer indicates. "a fluid with density $\rho(x;y;z;t=0)$" will be replaced by "a fluid with density $\rho(x;y;z;t)$", since we do not refer specifically to the initial density at this point, so that our formulation causes confusion..

**3.1 On the resolution**

On point **2.**, it is true that "if this entire study were performed at 1 or 0.1 km resolution the error measures would change". Even though we did not understand the exact procedure suggested by the Reviewer, we did perform the same simulation set at 2 km and 1 km resolutions. We provide here in Table 1 the equivalent

[Figure]

Figure 1: Ozone mixing ratio at $\frac{T}{2}$ as simulated in (a) the Van Leer simulation; (b) the PPM simulation, (c) the PPM+W simulation and (d) the Base simulation.

[Figure]

(a) TRC

(b) $O_3$

(c) NO

(d) $NO_2$

Figure 2: Time series for $\mathcal{E}_1$ ($\|\cdot\|_1$ normalized signature error) **the 1 km resolution simulation** for (a) TRC; (b) $O_3$; (c) NO and (d) $NO_2$

[Figure]

Figure 3: Minimum and maximum values in the 4 km simulations (presented in the manuscript) for: $\alpha_{CO} + \alpha_{CO_2}$ (panel (a)) and $\alpha_{NO} + \alpha_{NO_2} + \alpha_{HNO_3} + \alpha_{TRCb}$ (panel (b))

of Table 4 in the manuscript, for the same simulation but with 1 km resolution instead of 4 km, keeping the same Courant number (the time step is therefore also divided by 4 compared to the 4 km simulation, from 1800 s to 450 s).

The results are relatively consistent with the results shown in the manuscript, even though at this high resolution the performance of PPM improves relative to the Walcek scheme. At 4 km resolution, the PPM+W scheme was performing best for all species, but at 1 km the PPM scheme performs comparably for inert tracer TRC, and better for $NO_2$, while PPM+W performs better for $O_2$ and NO. Also, while at 4 km resolution the Walcek scheme was performing better than PPM for all species except $O_3$, the performance of PPM seems to improve faster than that of the Walcek scheme with increased resolution, consistently with the third-order design of the PPM scheme compared to second-order for the Walcek scheme.

While at 4 km resolution the PPM+W scheme was clearly and consistently outperforming all the other tested schemes, this is not the case at 1 km resolution, where it is clearly outperformed by the classical PPM scheme for one variable. However, even at this 1km resolution, the PPM+W scheme has the most consistent and robust performance, being either first or second in the comparison for all the variables and criteria.

Fig. 2 is the equivalent of Fig. 9 in the manuscript but with $\delta x = 1$ km, and shows the evolution in time of the $\mathcal{E}_1$ error in the various simulations for TRC, NO, $NO_2$ and $O_3$ respectively. It gives a vision of the time evolution of error along the run, confirming the extremely similar behaviour of PPM+W and PPM with regards to inert tracer TRC throughout the simulation (Fig. 2a), the equivalent or slightly better performance of PPM+W compared to PPM regarding NO throughout the run, and the better performance of PPM+W on $O_3$ throughout the run. The time series regarding $NO_2$ is more interesting, since we see that during the first half of the simulation the PPM+W run performs similarly to PPM, but in the second half the error increases faster, bringing PPM+W at error levels close to those of the Van Leer and Walcek schemes for this variable.

**These additional results for higher resolution will be commented in the revised version if we are invited to submit one.**

**3.2 On the conservation of sum species**

On point **3.**, we do not think this problem is specific to the Walcek time scheme. Apart from the Godunov donor-cell model, all other advection schemes in our study are non-linear. This is linked to the Godunov theorem, which states that there exists no linear and monotonous order-2 advection scheme, so that all the order-2 (and higher) schemes used in practice (and in our study) are non-linear, due to the use of slopelimiters. With these nonlinear slope limiters, if two species initially have a uniform sum, this uniformity will be lost during the run. Assessing such effects with the sum $NO + NO_2 + HNO_3$ is difficult since the sum is not initially uniform (it follows the Gaussian shape of Fig. 2 of this manuscript). To assess the magnitude to these undesirable effects with the various advection schemes we use here, **we have performed the same simulations with a new tracer species TRCb** so that initially we have $\alpha_{NO} + \alpha_{NO_2} + \alpha_{HNO_3} + \alpha_{TRCb} = 110$ ppb. We present the maps of $\alpha_{NO} + \alpha_{NO_2} + \alpha_{HNO_3} + \alpha_{TRCb}$ at the end of the experiment, and the time-series of the minimum and maximum of $\alpha_{NO} + \alpha_{NO_2} + \alpha_{HNO_3} + \alpha_{TRCb} = 110$ ppb. We also present the time series for $\alpha_{CO} + \alpha_{CO_2}$ which, in our study, should be constant, uniform and equal to 500 ppb everywhere in an exact solution, but is not for the same non-linearity reasons as with nitrogen.

Several elements can be inferred from Figure 3a-b:

- As expected, the linearity of the Godunov donor-cell scheme allows it to preserve exactly the uniform sum of two or several species with non-uniform distribution. In conformity with the Godunov theorem, higher-order schemes do not.

- Departure of the modelled sum compared to its expected values are up to $\pm 3\%$ for C and up to $\pm 3\%$ for N.

- Apart from the Godunov scheme, the Van Leer scheme is the least affected by this phenomenon, while the Walcek and PPM+W schemes are the most affected.

To verify that these discrepancies are indeed due to the treatment of the extrema, we provide maps of $\alpha_{NO} + \alpha_{NO_2} + \alpha_{HNO_3} + \alpha_{TRCb}$ at the end of the simulation (Fig. 4). These maps show that the discrepancies are indeed stronger in the vicinity of the peak of $NO_x$ mixing ratios, and that the error patterns are extremely similar in all schemes.

**Fig. 3 will be included in a revised version of this manuscript if we are invited to submit one**.

With our best regards,

S. Mailler, R. Pennel, L. Menut, A. Cholakian

[Figure]

Figure 4: $\alpha_{\mathrm{NO}} + \alpha_{\mathrm{NO_2}} + \alpha_{\mathrm{HNO_3}} + \alpha_{\mathrm{TRCb}}$ at $t = T$ as simulated in (a) the Van Leer simulation; (b) the PPM simulation, (c) the PPM+W simulation and (d) the Walcek simulation.

---

## Author Comment (AC2)

Answer to RC2 Anonymous Reviewer#2 10.5194/gmd-2023-78-RC2 on the manuscript
**"An improved version of the Piecewise Parabolic Method advection scheme: description and performance assessment in a bidimensional testcase with stiff chemistry in toyCTM v1.0."**

S. Mailler, R. Pennel, L. Menut, A. Cholakian

Oct. 2023

We are grateful to Anonymous Reviewer#2 for their careful reading and insightful comments on our manuscript. The main comments by Anonymous Reviewer#2 regard the need...

**Contents**

**1 Comments by Anonymous Reviewer#2**

**1.1 General comments**

The authors integrated PPM with the flux adjustment of Walcek method to improve the representation of local extrema which results in overall improvement without an increase in computational intensity. To evaluate their method they performed a simulation with different advection schemes with a chemical mechanism. They introduced the signature function as an alternative error statistic which basically evaluates whether the concentration probability distribution remains unchanged after advection. I think the advection scheme they suggested here is certainly an advancement for chemical transport modeling but the way they evaluated the method is in question.

The chemical transport model deals with the various velocity fields and initial conditions. I don't think the authors need to test their model with very realistic data, but just one simulation is not sufficient. For example, the key advancement of their method is on how to address local maxima, but there is only one local maxima in their initial condition. I'm curious how much the method will be effective if there are more stiff gradients–even testing with those scenarios can give them more benefits in computational cost.

They alleged that theoretically, the signature function will remain the same if there is no numerical error. I think they can change if the wind field has divergence. For example, if there is a negative divergence in a certain region, the concentration can be accumulated and the number of cells with high concentration can increase and the maximum range can increase. In case of the positive divergence, the wind will make scalars dispersed away so the number of cells with low concentration increases and the number of high concentration cells will decrease. The negative divergence scenario is relevant to high pollution episodes trapped by temperature inversion like LA smog while the positive divergence scenario is like the radial spread of volcanic ash. Even though those might be a bit extreme cases, it is the role of a chemical transport model to address those cases and in this regard the signature function cannot guarantee a proper evaluation.

If authors admit that the signature function works well on non-divergent wind scenarios including their simulation here, then it can make sense. However, the assertion that the function as a good method working universally is not true in my opinion.

**1.2 Minor details**

Below are minor details authors can correct:

L75 typo: . . . the three reactions wthat . . .

L80-90: R3 and R8 are incorrect in stoichiometry

Table 1: mean with standard deviation will be better to represent execution time

L256: The variable X should be explained.

Equation 11: H is non-italic here but in L259 it's italic.

L264: X is non-italic here.

L300: This sentence does not make sense to my perspective as I explained above.

L320: For similar reason, I don't think the maximum or minimum will not change. It is more natural to evolve with time when there is divergence.

L334: There might need a period in the middle. Looks like two different sentences are not properly separated.

L361: berformance looks like a typo of performance

L364 365: The conclusions drawn from the analysis. . . but the evidence is only a single experiment?

L375: Is a signature function really invariant? (same for other invariants authors mentioned)

**2 Answer to the General comments**

**Anonymous Reviewer#2** : "The chemical transport model deals with the various velocity fields and initial conditions. I don't think the authors need to test their model with very realistic data, but just one simulation is not sufficient. For example, the key advancement of their method is on how to address local maxima, but there is only one local maxima in their initial condition. I'm curious how much the method will be effective if there are more stiff gradients–even testing with those scenarios can give them more benefits in computational cost."

**Answer**: While it is true that only one local maximum exists in the initial condition (for tracers and nitrogen oxides), this is a testcase with active chemistry. And with the chemical evolution of the system, other configurations appear along the test for species with active chemistry. For example, ozone evolves into an absolute minimum of mixing ratio surrounded by a thin belt of high values then decreasing towards te background value away from the plume (Fig. 6 of the manuscript (reproduced here as Fig. 1). The initally bell-shaped maximum for inert tracers also evolves into a curved, elongated shape (Fig. 5 in the manuscript). The chemical evolution of concentrations does generate very stiff gradients. For example, Fig. 1 clearly shows that this high-ozone belt has a cross-dimension of the order of magnitude of one single grid-cell, which is extremely challenging for the advection framework.

**Anonymous Reviewer#2** : They alleged that theoretically, the signature function will remain the same if there is no numerical error. I think they can change if the wind field has divergence. For example, if there is a negative divergence in a certain region, the concentration can be accumulated and the number of cells with high concentration can increase and the maximum range can increase. In case of the positive divergence, the wind will make scalars dispersed away so the number of cells with low concentration increases and the number of high concentration cells will decrease. The negative divergence scenario is relevant to high pollution episodes trapped by temperature inversion like LA smog while the positive divergence scenario is like the radial spread of volcanic ash. Even though those might be a bit extreme cases, it is the role of a chemical transport model to address those cases and in this regard the signature function cannot guarantee a proper evaluation.

**Answer**: We maintain that, in the absence of mixing, the signature function is strictly preserved by the

[Figure]

Figure 1: Ozone mixing ratio at $\frac{T}{2}$ as simulated in (a) the Van Leer simulation; (b) the PPM simulation, (c) the PPM+W simulation and (d) the Base simulation.

advection equation–Eq. 10 in the initial manuscript, reproduced here as Equation. 1:

$$\frac{\partial \alpha}{\partial t} + \mathrm{u}\nabla\alpha = 0. \tag{1}$$

The proof for this is given from lines 264-269 of the manuscript along with Eq. 11, which formally defines the signature function–reproduced here as Eq. 2:

$$\mathcal{S}^t(X) = \frac{\int_{\mathcal{D}} \mathrm{H}\left(X - \alpha\left(x; y; z; t\right)\right)\rho\mathrm{d}\mathcal{V}}{\int_{\mathcal{D}}\rho\mathrm{d}\mathcal{V}}, \tag{2}$$

. It is true that, as the Reviewer says, the convergent (resp. divergent) character of the wind field will increase (resp. decrease) the *concentration* of an inert tracer tracer along a trajectory, but it will leave inchanged its *mixing ratio*, which is the key point in constructing the signature function and showing that it is, indeed, invariant under the action of pure advection.

Equation 2, defining the signature function $X \mapsto \mathcal{S}(X)$, is constructed as a ratio. Both the numerator and the denominator of this ratio are constant in time due to the specific properties of the advection equation and the continuity equation (Eq. 9 in the manuscript), and this holds even if the wind field has a divergence:

- Numerator, $\int_{\mathcal{D}} \mathrm{H}\left(X - \alpha\left(x; y; z; t\right)\right)\rho\mathrm{d}\mathcal{V}$, represents the mass of fluid contained in fluid parcels with tracer mixing ratio $\alpha\left(x; y; z; t\right) < X$. THis is constant in time (for any value of $X$) because the mass of any fluid parcel is constant in time (this is a property of the continuity equation), and the tracer mixing ratio in any fluid parcel is also constant in time (this is a property of the advection equation, Eq. 1).

- Denominator, $\int_{\mathcal{D}}\rho\mathrm{d}\mathcal{V}$, represents the total mass of fluid (e.g., air), which is preserved if there is no flux across the domain boundary, which is essentially the case of, e.g., the entire atmosphere, or if the fluid moves within a "box" as in our academic case.

Therefore, the signature function $\mathcal{S}$ is rigorously invariant under the action of the advection equation, even in the case in which the wind field is divergent.

This is true of the complete, formal definition shown in Eq. 11. in the manuscript. Eq. 12 shows how this definition can be discretitzed to test the properties of numerical advection schemes in a Eulerian model without loss of generality (including the case of a divergent wind field). Equations 13-14 in the manuscript is just a practical "trick" to calculate very easily the signature error in the particular case where every cell of the model contains the same mass of fluid, and this mass of fluid does not vary in time. While is is already said in the manuscript that this "trick" applies only "in the particular case in which the carrier fluid mass $\rho_i V_i$ is the same in all model cells", which is true throughout time only for non-divergent fluxes.
**✓ the comment by Anonymous Reviewer#2 shows that a clarification of which quantities are invariant even in divergent fluxes, and which quantities are not would be helpful in the text. Explicit statements in this sense have been included in Section 4.2..**

**Anonymous Reviewer#2** : If authors admit that the signature function works well on non-divergent wind scenarios including their simulation here, then it can make sense. However, the assertion that the function as a good method working universally is not true in my opinion.
**Answer**: We do not assert that the signature function is working universally. However, in the conclusion, we will precise the range of applicability of the invariance of the signature function for realistic applications. While, as discussed above, the invariance of function $\mathcal{S}$ holds in the presence of divergence, it applies only to pure advection problems, excluding the presence of diffusion and mixing.
**✓ These precisions are brought into the conclusion in the revised version of the manuscript in order to better specify and discuss the possible applicability of the signature function as an invariant in realistic problems.**

**3  Specific answers**

**Anonymous Reviewer#2** :L75 typo: . . . the three reactions wthat . . .

✓ Corrected

**Anonymous Reviewer#2** :L80-90: R3 and R8 are incorrect in stoichiometry

**Answer**: The fact that R8 is not balanced in stoechiometry is due to the omission of $O_2$ as a reactant, as explained in lines 220-223 in the manuscript. The imbalance it introduces in the system is only on dioxygen, it is common practice in atmospheric chemistry to ignore such imbalances because they do not affect measurably the amount of oxygen available. Introducing this formally imbalanced reaction saves computation time by avoiding to introduce atomic hydrogen H as an intermediary species.

The incorrect stoechiometry in reaction R3 is also due to the omission of $O_2$, this time as a product. This is also common practice in chemistry-transport modelling for the same reason as above, but ✓ **consistently with reactions R4, R7, R10 and R11, $O_2$ will be included in the products list in the revised version**.

**Anonymous Reviewer#2** : Table 1: mean with standard deviation will be better to represent execution time

**Answer**: As shown by Table 1, the execution time for one call to the advection routine is a few dozen nanoseconds at max. the system clock we could acess for this benchmark has a time step of 1 ms, which is why we have run each of the schemes $1.04 \times 10^8$ times (line 183) to evaluate the calculation time for one call. With this method, we are able to evaluate the average time but not its distribution, and we do not see a way to go beyond this limit. The initial caption of Table 1 may suggest that the exectution time for the schemes is constant regardless of the input values, which is not the case due to the conditional statements in the schemes. ✓ **In the revised version, we make clear that the value we give is an average by writing "Mean calculation time" instead of "Calculation time".**

**Anonymous Reviewer#2** :L256: The variable X should be explained.

**Answer**: X is a dummy variable used only to define function $\mathcal{S}$. ✓ **we have clarified the meaning of X in the revised version:** For any given time $t$ and any mixing ratio $0 \leq X \leq 1$, we can define $\mathcal{S}^t(X)$ ...

**Anonymous Reviewer#2** : Equation 11: H is non-italic here but in L259 it's italic.

**Answer**: ✓ **Fixed, thanks**

**Anonymous Reviewer#2** :L264: X is non-italic here.

**Answer**: ✓ **Fixed, thanks**

**Anonymous Reviewer#2** L300: This sentence does not make sense to my perspective as I explained above.

**Answer**: We are not able to understand this concern, and do not see any reference to this is the above comments, sorry.

**Anonymous Reviewer#2** L320: For similar reason, I don't think the maximum or minimum will not change. It is more natural to evolve with time when there is divergence.

**Answer**: The minimum and maximum of the mixing ratio (not of the concentration) have to stay constant in time even in the presence of divergence, *per* Eq. 1 which shows that the mixing ratio has to stay constant along any trajectory (this equation holds even if the wind field is divergent).

**Anonymous Reviewer#2** L334: There might need a period in the middle. Looks like two different sentences are not properly separated.

**Answer**: there was a point missing in the middle, ✓ **it has been fixed, thank you.**

**Anonymous Reviewer#2** L361: berformance looks like a typo of performance

**Answer**: ✓ **it has been fixed, thank you.**

**Anonymous Reviewer#2** L364 365: The conclusions drawn from the analysis... but the evidence is only a single experiment?

✓ We have added to the conclusion additional discussion on the relevance and possible limits of our conclusions, including the fact that they rely mostly on one test-case. Also, the revised version presents more numerical experiments to mitigate this limit, including a convergence test and a higher-resolution simulation.

**Anonymous Reviewer#2** L375: Is a signature function really invariant? (same for other invariants authors mentioned)

**Answer** As discussed above, we feel that we have proved that the signature function is really an invariant

(see the discussion above). Also, the minimum and maximum of tracer mixing ratio are indeed invariants in the advection equation (unlike maxima and minima of tracer density) as discussed in, e.g., Brasseur and Jacob [2017] : "For a constant velocity, the advection of both the density and the mixing ratio is represented by a simple translation (without deformation) of the initial function in the direction of the velocity. If the velocity decreases with space, the initial distribution of both quantities is distorted as the material is advected. The value of the maximum mixing ratio is unchanged, but the maximum value of the density is enhanced. Advection can thus modify extrema of tracer densities in a diverging flow."

S. Mailler, R. Pennel, L. Menut, A. Cholakian

**References**

G. P. Brasseur and D. J Jacob. Chap. 7, numerical methods for advection. In *Modeling of atmospheric chemistry*. Cambridge University Press, 2017.

---

## Author Comment (AC3)

Answer to Dr. Hilary Weller 10.5194/gmd-2023-78-RC3 on the manuscript
**"An improved version of the Piecewise Parabolic Method advection scheme: description and performance assessment in a bidimensional testcase with stiff chemistry in toyCTM v1.0."**

S. Mailler, R. Pennel, L. Menut, A. Cholakian

Oct. 2023

We are grateful to Dr. Wellerfor her careful reading and insightful comments on our manuscript.

**Contents**

**1 Comments by Dr. Weller**

**1.1 General comments**

This is a very nice paper that describes evaluation of tracer transport using a signature function, as well as the improved PPM scheme mentioned in the title. The signature function gives a lot of information about the evolution of transport errors without the need for an analytic solution. I like it and I am surprised that it is not already widely used.

I have two main comments which the authors may argue is beyond the scope, but these comments, I think, would make the paper even more convincing.

**1.** A new advection scheme is introduced - PPM+W. The results of your test cases look good but you are comparing against some old advection schemes using a non-standard test case. Please include results using a standard test case so that you can refer to other papers that include results of exactly the same test cases and the reader can easily compare errors.

**2.** You propose a new test case. There isn't enough evidence presented to demonstrate that a new test case is needed.

**1.2 Minor comments**

**1.** "Non-monotonic" rather than "non-monotonous".

**2.** The section heading 2.2 "Flux description" is not well named. This is really "Definition of Test Case".

**3.** It is not clear why the test case described in sections 2.2.1-2.3 is not exactly the same as LeVeque (1996). Using dimensioned rather than non-dimensional variables does not constitute an effective change.

**4.** Line 147. Replace with "PPM (Colella and Woodward, 1984)" as that is how you refer to it elsewhere.

**5.** The errors of van Leer and PPM are not obvious in figures 3 and 4. These schemes look better than the +W schemes. Would it be possible or helpful to compare the scheme cell averages with the exact cell averages?

**6.** Is it usual for chemistry schemes to violate machine precision conservation? Provide references.

**7.** You often start a new paragraph where there shouldn't be a new paragraph. Eg line 224 and 259. Note that any blank line in LaTeX will create a new paragraph. So don't put blank lines around equations unless you want a new paragraph.

**8.** Equations 9 and 10 need u to be displayed as a vector and equation 10 needs a dot between u and grad.

**2 Answer to the General comments**

We are very grateful of the consideration and positive appreciation given by Dr. Weller on our work.

**1.** "A new advection scheme is introduced - PPM+W. The results of your test cases look good but you are comparing against some old advection schemes using a non-standard test case. Please include results using a standard test case so that you can refer to other papers that include results of exactly the same test cases and the reader can easily compare errors."

**Answer** We agree that much work has been done on advection schemes since the design of the Van Leer [1977], Colella and Woodward [1984] and Walcek [2000] schemes. In this sense it is fair to say that we compare PPM+W to "some old advection schemes". This is why we are not able to claim (and we don't) that the PPM+W scheme represents an improvement over the latest state-of-the art regarding advection schemes. Most likely, it doesn't. However, as we argue, the PPM scheme [Colella and Woodward, 1984] is widely used in state-of-the-art geophysical models. Such models include oceanic models – see Gibson et al. [2017], Dietze et al. [2020], air quality models includeing the very popular models CMAQ, CAMx and Geos-CHEM. Atmospheric models such as GFDL [Harris et al., 2021] and Meso-NH [Lac et al., 2018] also use PPM.

This widespread use of the PPM scheme is probably due to the fact that, in the wording of Harris et al. [2021], it can be considered as "highly accurate and efficient enough to be useful", meaning that for some geophysical application it represents a good compromise between accuracy and computational cost. The recent paper of Cao et al. [2023] shows how PPM is being adapted to the latest technological evolution in order to optimize its good numerical efficiency in new architectures.

As a summary regarding PPM, while we are aware of the existence of higher-order or alternative approaches with improved accuracy, we think that PPM is still very used as a good and robust compromise between accuracy and numerical precision, so that any improvement to this scheme (in our case, improving accuracy without degrading efficiency) is potentially relevant, not as a novelty in the quest for more accurate advection schemes, but as a possible practical improvement in the fields of geophysics where PPM seems to be one of the best compromises between accuracy and cost, and PPM+W might further improve this balance by improving accuracy without increasing the cost.

We agree that the use of the other schemes we present as benchmarks, Van Leer [1977] and Walcek [2000] have become less frequent nowadays. Both these schemes are among the available options in the CHIMERE chemistry-transport model along with PPM [Menut et al., 2021], and Walcek is the advection scheme retained in the LOTOS-EUROS chemistry-transport model [Timmermans et al., 2022].

✓ **In the introduction of the revised version we will include references to higher-order schemes, with Prather [1986] and Waruszewski et al. [2018] as examples, and mention explicitly the fact that PPM is hconsidered as a good balance between accuracy and computational efficiency–Harris et al. [2021].**

Regarding the choice of the test case, we have worked with the swirling rotational flow exactly as defined in LeVeque [1996], we do not know if this can be qualified as a "standard" test case, most standard test cases we know of are designed on the sphere [Lauritzen et al., 2014], which is not adapted for applications to regional models. One of the test cases presented by Lauritzen is adapted from the LeVeque [1996] swirling rotational flow, which we decided to use for our study. It has several advantages that suit our purpose:

- The flow is non-constant in time which makes it both more realistic (compared to the real conditions in a chemistry-transport model), and a more challenging test of the correctness of the implementation.

- The flow is sheared, making it challenging by stretching the plumes into filaments.

- The flow is contained within a square box, preventing the need for a boundary condition and allowing to check mass conservation (unlike, *e.g.*, solid rotation).

The LeVeque [1996] study itself does not provide quantitative results in terms of reproducible metrics. More generally, we did not find a suitable test-case with a simplified but stiff chemistry in a square area (well-suited to the test of the advections schems tested here, which are designed for cartesian geometry).

Producing another test-case as suggested would have made the study very tedious. However, in the revised version, we have made two improvements in this direction:

- ✓ we perform a convergence test to compare the selected advection schemes across a range of resolutions, confirming that PPM+W performs consistently better than PPM even when resolution increases (Section 3.1.4, Fig. 5 and Table 2 in the revised version);

- ✓ we provide results from a higher resolution simulation of the same test case (Table 6 in the revised version). These results confirm a generally better performance of PPM+W compared to other schemes, but the edge over PPM is more reduced and one species is better reproduced by PPM at this high resolution.

Also, in the conclusion of the revised version, we insist more on the fact that we have not tested this new scheme in realistic conditions, and that we have not tested the full range of Courant numbers and of possible tracer patterns.

**2.** You propose a new test case. There isn't enough evidence presented to demonstrate that a new test case is needed.

See above, we did not find a standard test case designed for a chemistry-transport problem in cartesian geometry, which is why we have designed our own test case build a ultra-simplified –but stiff– system representing tropospheric chemistry, and a flow (which is exactly the swirling deformational flow of LeVeque [1996]), that has the desirable properties listed above.

The initial formulation ("This flow is defined from the ideas of LeVeque [1996].") was leading the reader to understand that we have defined a new flow, that would be "almost" the swirling deformation flow of LeVeque [1996], while we actually used *the exact flow* of LeVeque [1996].

✓ **in the revised version, this sentence has been changed to:**

The flow we use in this study is the swirling deformational flow introduced by LeVeque [1996] (their Eqs. 9.5-9.6): . . .

**3   Answers to the Minor comments**

**1.** "Non-monotonic" rather than "non-monotonous".

✓ . We have also replaced two more occurences of "monotonous" by "monotonic'

**2.** The section heading 2.2 "Flux description" is not well named. This is really "Definition of Test Case".

✓ . The section has been renamed as suggested.

**3.** It is not clear why the test case described in sections 2.2.1-2.3 is not exactly the same as LeVeque (1996). Using dimensioned rather than non-dimensional variables does not constitute an effective change.

✓ The flow is actually exactly the same as LeVeque [1996] (but with dimensioned variables). The fact that this is exactly the same flow has been clarified in the revised version..

**4.** Line 147. Replace with "PPM (Colella and Woodward, 1984)" as that is how you refer to it elsewhere.

✓ THis has been changed accordingly.

**5.** The errors of van Leer and PPM are not obvious in figures 3 and 4. These schemes look better than the +W schemes. Would it be possible or helpful to compare the scheme cell averages with the exact cell averages?

The Van Leer and PPM schemes look better than the "+W" schemes in this figure because they are an actual polynomial (degree-1 for Van Leer and degree-2 for PPM) reconstruction of the exact flow, which the Walcek-corrected slopes are not. As said in the manusript, the idea of the Walcek slope modifications is to intentionnally overestimate the fluxes into the cell with the maximum mixing ratio in order to better maintain this maximum value. Therefore, the Walcek reconstruction departs more strongly from the exact mixing ratio than the Van Leer or PPM schemes because it is not a polynomial reconstruction of the exact mixing ratio. Hawever, as shown by Walcek [2000], this intentional departure from the polynomial reconstruction brings added skill to the scheme in resolving the advection equation.

It would not be helpful to compare the cell averages with the exact cell averages, becaus in all gthese schemes, by construction, the cell-average of the reconstructed concentration is equal to the exact cell average.

**6.** Is it usual for chemistry schemes to violate machine precision conservation? Provide references.

This is usual, unfortunately. It is not frequent that authors discuss this problem explicitly in their publications. For example, Brasseur and Jacob [2017] in their chapter 6 on numerical methods for chemical systems only allude very briefly to this issue ("Mass is not fully conserved by the Jacobi and Gauss–Seidel iterative procedures"). Cariolle et al. [2017] (cited line 197 of the initial manuscript) does discuss this issue in some detail however in their Section 2.1, giving more precision on the reason for this lack of mass conservation: the scheme itself (Eq. 6 in our initial manuscript) is mass conservative, however "the mass conservation can only be obtained if a good convergence of the solution is reached".

✓ We have added a new reference to Cariolle et al. [2017] at the point when we explain why perfect mass conservation is not obtained

**7.** You often start a new paragraph where there shouldn't be a new paragraph. Eg line 224 and 259. Note that any blank line in LaTeX will create a new paragraph. So don't put blank lines around equations unless you want a new paragraph.

✓ We have checked and corrected these undesirable new paragraphs around equations, we hope there is none left in the revised version.

**8.** Equations 9 and 10 need u to be displayed as a vector and equation 10 needs a dot between u and grad.

✓ This has been fixed, thanks.

S. Mailler, R. Pennel, L. Menut, A. Cholakian

**References**

G. P. Brasseur and D. J Jacob. *Modeling of atmospheric chemistry*. Cambridge University Press, 2017. doi: 10.1017/9781316544754.

K. Cao, Q. Wu, L. Wang, N. Wang, H. Cheng, X. Tang, D. Li, and L. Wang. Gpu-hadvppm v1.0: a high-efficiency parallel gpu design of the piecewise parabolic method (ppm) for horizontal advection in an air quality model (camx v6.10). *Geoscientific Model Development*, 16(15):4367–4383, 2023. doi: 10.5194/gmd-16-4367-2023.

D. Cariolle, P. Moinat, H. Teyssèdre, L. Giraud, B. Josse, and F. Lefèvre. Asis v1.0: an adaptive solver for the simulation of atmospheric chemistry. *Geosci. Model Dev.*, 10(4):1467–1485, 2017. doi: 10.5194/gmd-10-1467-2017.

P. Colella and P. R. Woodward. The piecewise parabolic method (PPM) for gas-dynamical simulations. *J. Comput. Phys.*, 11:38–39, 1984. doi: 10.1016/0021-9991(84)90143-8.

H. Dietze, U. Löptien, and J. Getzlaff. Momso 1.0 – an eddying southern ocean model configuration with fairly equilibrated natural carbon. *Geosci. Model Dev.*, 13(1):71–97, 2020. doi: 10.5194/gmd-13-71-2020.

Angus H. Gibson, Andrew McC. Hogg, Andrew E. Kiss, Callum J. Shakespeare, and Alistair Adcroft. Attribution of horizontal and vertical contributions to spurious mixing in an arbitrary lagrangian–eulerian ocean model. *Ocean Modelling*, 119:45–56, 2017. ISSN 1463-5003. doi: https://doi.org/10.1016/j.ocemod.2017.09.008. URL https://www.sciencedirect.com/science/article/pii/S1463500317301440.

Lucas Harris, Xi Chen, William Putman, and Jan-Huey Zhou, Linjiong andChen. A scientific description of the gfdl finite-volume cubed-sphere dynamical core. Technical report, United States. National Oceanic and Atmospheric Administration. Office of Oceanic and Atmospheric Research. ; Geophysical Fluid Dynamics Laboratory (U.S.), 2021.

C. Lac, J.-P. Chaboureau, V. Masson, J.-P. Pinty, P. Tulet, J. Escobar, M. Leriche, C. Barthe, B. Aouizerats, C. Augros, P. Aumond, F. Auguste, P. Bechtold, S. Berthet, S. Bielli, F. Bosseur, O. Caumont, J.-M. Cohard, J. Colin, F. Couvreux, J. Cuxart, G. Delautier, T. Dauhut, V. Ducrocq, J.-B. Filippi, D. Gazen,

O. Geoffroy, F. Gheusi, R. Honnert, J.-P. Lafore, C. Lebeaupin Brossier, Q. Libois, T. Lunet, C. Mari, T. Maric, P. Mascart, M. Mogé, G. Molinié, O. Nuissier, F. Pantillon, P. Peyrillé, J. Pergaud, E. Perraud, J. Pianezze, J.-L. Redelsperger, D. Ricard, E. Richard, S. Riette, Q. Rodier, R. Schoetter, L. Seyfried, J. Stein, K. Suhre, M. Taufour, O. Thouron, S. Turner, A. Verrelle, B. Vié, F. Visentin, V. Vionnet, and P. Wautelet. Overview of the meso-nh model version 5.4 and its applications. *Geosci. Model Dev.*, 11(5): 1929–1969, 2018. doi: 10.5194/gmd-11-1929-2018.

P. H. Lauritzen, P. A. Ullrich, C. Jablonowski, P. A. Bosler, D. Calhoun, A. J. Conley, T. Enomoto, L. Dong, S. Dubey, O. Guba, A. B. Hansen, E. Kaas, J. Kent, J.-F. Lamarque, M. J. Prather, D. Reinert, V. V. Shashkin, W. C. Skamarock, B. Sørensen, M. A. Taylor, and M. A. Tolstykh. A standard test case suite for two-dimensional linear transport on the sphere: results from a collection of state-of-the-art schemes. *Geosci. Model Dev.*, 7(1):105–145, 2014. doi: 10.5194/gmd-7-105-2014.

Randall J. LeVeque. High-resolution conservative algorithms for advection in incompressible flow. *SIAM J. Numer. Anal.*, 33(2):627–665, 1996. doi: 10.1137/0733033.

L. Menut, B. Bessagnet, R. Briant, A. Cholakian, F. Couvidat, S. Mailler, R. Pennel, G. Siour, P. Tuccella, S. Turquety, and M. Valari. The chimere v2020r1 online chemistry-transport model. *Geosci. Model Dev.*, 14(11):6781–6811, 2021. doi: 10.5194/gmd-14-6781-2021.

Michael J. Prather. Numerical advection by conservation of second-order moments. *J. Geophys. Res.–Atmos.*, 91(D6):6671–6681, 1986. doi: 10.1029/JD091iD06p06671.

R. Timmermans, D. van Pinxteren, R. Kranenburg, C. Hendriks, K.W. Fomba, H. Herrmann, and M. Schaap. Evaluation of modelled lotos-euros with observational based pm10 source attribution. *Atmos. Environ.: X*, 14:100173, 2022. ISSN 2590-1621. doi: 10.1016/j.aeaoa.2022.100173.

B. Van Leer. Towards the ultimate conservative difference scheme. iv. a new approach to numerical convection. *J. Comput. Phys.*, 23(3):276 – 299, 1977. ISSN 0021-9991. doi: 10.1016/0021-9991(77)90095-X.

Chris J. Walcek. Minor flux adjustment near mixing ratio extremes for simplified yet highly accurate monotonic calculation of tracer advection. *J. Geophys. Res.*, 105(D7):9335–9348, 2000. doi: 10.1029/ 1999JD901142.

Maciej Waruszewski, Christian Kühnlein, Hanna Pawlowska, and Piotr K. Smolarkiewicz. Mpdata: Third-order accuracy for variable flows. *J. Comput. Phys.*, 359:361–379, 2018. ISSN 0021-9991. doi: 10.1016/j. jcp.2018.01.005.

---

## Author Comment (AC4)

Answer to the Reviewer Comment signed by Christopher Walcek 10.5194/gmd-2023-78-RC1 on the manuscript "**An improved version of the Piecewise Parabolic Method advection scheme: description and performance assessment in a bidimensional testcase with stiff chemistry in toyCTM v1.0.**"

S. Mailler, R. Pennel, L. Menut, A. Cholakian

July 28, 2023.

We are grateful to Christopher Walcek for his careful reading and insightful comments on our manuscript. Two main comments by Pr. Walcek regard the need to discuss how a higher-resolution would affect our results, and the question of the "inability to preserve lumped but conserved species". We address both these comments in detail below.

**Contents**

**1 Comments by C. Walcek**

The authors introduce a modification and improvement of numerical advection algorithms by merging existing techniques in a way the optimizes the numerical approximation of advection. They also introduce a new error metric that assesses not only local extrema, but also the distribution and range of concentration distributions that are advected. My main complaint about the study is the limited nature of the test presented (only one Courant number, and one fairly well-resolved gaussian hill shape. Aside from a few minor pounts that need to be mentioned, the paper is publishable.

**1.** typo line 250. How can fluid density everywhere be zero. is this a typo ?

**2.** using these deformational flows, figures like 8c and others do not properly quantify the true "accuracy" of the scheme since the shape is advected/sheared into a shape that has scale features that are smaller than the 4 km resolution of this experiment. If this entire study were performed at 1 or 0.1 km resolution, the error measures would change. Basically, the simulation called "base" in these figures is not a true average of the 4x4km resolution "base". When the peak of the "base" Guassuan shape is sheared into successively smaller and smaller cells, then averaged BACK to the 4 km coarse T=0 4km-resolution discrete grid, then the simulation called "base" should change with time also. This applies to the figures which show the distribution at the T/2 times at the point where the. Here the authors could do a simulation at spatial resolution of 0.4 km (much smaller than 4 km), then MAP the 0.4 km simulation back onto the 4km "base" grid by averaging 10X10 grid cells into a single 4 km grid, then compare.

**3.** one of the problems with the Walcek peak scheme is the inability to preserve lumped but conserved species since there it is guaranteed that numerical treatment around local maxima are NOT treated identical in SUMS of conserved species. The "peaks" in NO, $NO_2$ and $HNO_3$ will occur at different places than the combined $NO+NO_2+HNO_3$. In the context of the problem presented here, total nitrogen should be conserved (NO+NO2+HNO3). The authors should be able to show that there is non-monotonic behavior of any advection scheme which treats local EXTREMES algebraically differently. Please show graphs of $NO+NO_2+HNO_3$.

**2 General answer**

Pr. Walcek judges this paper publishable, with one main complaint about "the limited nature of the test presented (only one Courant number, and one fairly well-resolved gaussian hill shape)".

We agree that the fact that there is only one test presented could be a limit, however the manuscript presents objective scores for two different metrics and 4 different species on a total of 5 simulations, amounting to 40 metric scores, and these different species test the advection schemes in different conditions because their distributions are very different. We also introduce one new advection scheme (that has to be explained) and a different metric with the idea, as Pr. Walcek describes it, to assess "not only local extrema, but also the distribution and range of concentration distributions that are advected". This has guided the choice to perform only one test-case (but with several advection configurations and chemical species).

It is true that there is "only one Courant number", which does evolve in time though since the mass-flux (and therefore the CFL) is equal to zero at $\frac{T}{2}$. For the same reasons as above, we did not perform a sensitivity test on timestep.

✓ **These are acknowledged in the conclusion of the revised manuscript. Also, to slightly generalize and strengthen our results without performing a full-fledge new test-case exêriment, we have also performed a 1d convergence test for all the studied advection schemes with an inert tracer (Section 3.1.4, Figure 5 and Table 2). This confirms our results, and shows thaht the improvement edge of PPM+W over PPM persists throughout all resolutions.**

Regarding the fact that our test case consists essentially of a "fairly well-resolved gaussian hill shape", this is essentially true for the inert tracers, even at half-time under the influence of wind shear (Fig. 5 of the manuscript), but this is not true for the ozone concentrations. Fig. 6 of the manuscript (reproduced here as Fig. 1) shows that the high-ozone belt that circles the plume takes an extremely elongated shape under the effect of wind shear and of the different chemical regimes. Fig. 1 clearly shows that this high-ozone belt has a cross-dimension of the order of magnitude of one single grid-cell, which is extremely challenging for the advection framework.

**3 Specific answers**

**3.1 On density**

On point **1.**, equation $\rho(x; y; z; t = 0)$ is not properly written, as the Reviewer indicates.

✓ **"a fluid with density $\rho(x; y; z; t = 0)$" will be replaced by "a fluid with density $\rho(x; y; z; t)$", since we do not refer specifically to the initial density at this point, so that our formulation causes confusion.**

**3.2 On the resolution**

On point **2.**, it is true that "if this entire study were performed at 1 or 0.1 km resolution the error measures would change". Even though we did not understand the exact procedure suggested by the Reviewer, we did perform the same simulation set at 2 km and 1 km resolutions. We provide here in Table 1 the equivalent of Table 4 in the manuscript, for the same simulation but with 1 km resolution instead of 4 km, keeping the same Courant number (the time step is therefore also divided by 4 compared to the 4 km simulation, from 1800 s to 450 s).

[Figure]

Figure 1: Ozone mixing ratio at $\frac{T}{2}$ as simulated in (a) the Van Leer simulation; (b) the PPM simulation, (c) the PPM+W simulation and (d) the Base simulation.

|  | Normalized $\|\cdot\|_1$ error | | | | $\mathcal{E}_1$ | | | |
|---|---|---|---|---|---|---|---|---|
|  | TRC | O3 | NO2 | NO | TRC | O3 | NO2 | NO |
| Godunov | 0.434 | 0.275 | 0.357 | 0.584 | 0.386 | 0.244 | 0.237 | 0.564 |
| Van Leer | 0.0417 | 0.0486 | 0.0521 | 0.0905 | 0.0241 | 0.0418 | 0.0325 | 0.0773 |
| Walcek | 0.0202 | 0.0303 | 0.0423 | 0.0509 | 0.016 | 0.0244 | 0.0344 | 0.0472 |
| PPM | 0.0169 | 0.0336 | **0.0324** | 0.0574 | **0.00881** | 0.0221 | **0.0263** | 0.0382 |
| PPM+W | **0.0148** | **0.0247** | 0.0383 | **0.0483** | 0.00901 | **0.0112** | 0.031 | **0.0332** |

Table 1: Normalized $\|\cdot\|_1$ error $E_1$ and normalized $\|\cdot\|_1$ signature error $\mathcal{E}_1$ **in the 1 km simulations** at the end of the simulations for $O_3$, NO, $NO_2$ and TRC, compared to the Base simulation with no advection. In each column, the lowest error value is in bold font, the second-lowest in underlined.

[Figure]

(a) TRC

(b) $O_3$

(c) NO

(d) $NO_2$

Figure 2: Time series for $\mathcal{E}_1$ ($\|\cdot\|_1$ normalized signature error) **the 1 km resolution simulation** for (a) TRC; (b) $O_3$; (c) NO and (d) $NO_2$

[Figure]

Figure 3: Minimum and maximum values in the 4 km simulations (presented in the manuscript) for: $\alpha_{CO} + \alpha_{CO_2}$ (panel (a)) and $\alpha_{NO} + \alpha_{NO_2} + \alpha_{HNO_3} + \alpha_{TRCb}$ (panel (b))

The results are relatively consistent with the results shown in the manuscript, even though at this high resolution the performance of PPM improves relative to the Walcek scheme. At 4 km resolution, the PPM+W scheme was performing best for all species, but at 1 km the PPM scheme performs comparably for inert tracer TRC, and better for $NO_2$, while PPM+W performs better for $O_2$ and NO. Also, while at 4 km resolution the Walcek scheme was performing better than PPM for all species except $O_3$, the performance of PPM seems to improve faster than that of the Walcek scheme with increased resolution, consistently with the third-order design of the PPM scheme compared to second-order for the Walcek scheme.

While at 4 km resolution the PPM+W scheme was clearly and consistently outperforming all the other tested schemes, this is not the case at 1 km resolution, where it is clearly outperformed by the classical PPM scheme for one variable. However, even at this 1km resolution, the PPM+W scheme has the most consistent and robust performance, being either first or second in the comparison for all the variables and criteria.

Fig. 2 is the equivalent of Fig. 9 in the manuscript but with $\delta x = 1\,\text{km}$, and shows the evolution in time of the $\mathcal{E}_1$ error in the various simulations for TRC, NO, $NO_2$ and $O_3$ respectively. It gives a vision of the time evolution of error along the run, confirming the extremely similar behaviour of PPM+W and PPM with regards to inert tracer TRC throughout the simulation (Fig. 2a), the equivalent or slightly better performance of PPM+W compared to PPM regarding NO throughout the run, and the better performance of PPM+W on $O_3$ throughout the run. The time series regarding $NO_2$ is more interesting, since we see that during the first half of the simulation the PPM+W run performs similarly to PPM, but in the second half the error increases faster, bringing PPM+W at error levels close to those of the Van Leer and Walcek schemes for this variable.

✓ **Additional results for higher resolution are commented in the revised version (section 5.4 and Table 6), they also help discuss the possible improvement of PPM+W over PPM in the conclusions.**

**3.3 On the conservation of sum species**

On point **3.**, we do not think this problem is specific to the Walcek time scheme. Apart from the Godunov donor-cell model, all other advection schemes in our study are non-linear. This is linked to the Godunov theorem, which states that there exists no linear and monotonous order-2 advection scheme, so that all the order-2 (and higher) schemes used in practice (and in our study) are non-linear, due to the use of slope-limiters. With these nonlinear slope limiters, if two species initially have a uniform sum, this uniformity will be lost during the run. Assessing such effects with the sum $NO + NO_2 + HNO_3$ is difficult since the sum is

[Figure]

Figure 4: $\alpha_{\mathrm{NO}} + \alpha_{\mathrm{NO_2}} + \alpha_{\mathrm{HNO_3}} + \alpha_{\mathrm{TRCb}}$ at $t = T$ as simulated in (a) the Van Leer simulation; (b) the PPM simulation, (c) the PPM+W simulation and (d) the Walcek simulation.

not initially uniform (it follows the Gaussian shape of Fig. 2 of this manuscript). To assess the magnitude to these undesirable effects with the various advection schemes we use here, **we have performed the same simulations with a new tracer species TRCb** so that initially we have $\alpha_{\mathrm{NO}} + \alpha_{\mathrm{NO_2}} + \alpha_{\mathrm{HNO_3}} + \alpha_{\mathrm{TRCb}} = 110\,\mathrm{ppb}$. We present the maps of $\alpha_{\mathrm{NO}} + \alpha_{\mathrm{NO_2}} + \alpha_{\mathrm{HNO_3}} + \alpha_{\mathrm{TRCb}}$ at the end of the experiment, and the time-series of the minimum and maximum of $\alpha_{\mathrm{NO}} + \alpha_{\mathrm{NO_2}} + \alpha_{\mathrm{HNO_3}} + \alpha_{\mathrm{TRCb}} = 110\,\mathrm{ppb}$. We also present the time series for $\alpha_{\mathrm{CO}} + \alpha_{\mathrm{CO_2}}$ which, in our study, should be constant, uniform and equal to $500\,\mathrm{ppb}$ everywhere in an exact solution, but is not for the same non-linearity reasons as with nitrogen.

Several elements can be inferred from Figure 3a-b:

- As expected, the linearity of the Godunov donor-cell scheme allows it to preserve exactly the uniform sum of two or several species with non-uniform distribution. In conformity with the Godunov theorem, higher-order schemes do not.

- Departure of the modelled sum compared to its expected values are up to $\pm 3\%$ for C and up to $\pm 3\%$ for N.

- Apart from the Godunov scheme, the Van Leer scheme is the least affected by this phenomenon, while the Walcek and PPM+W schemes are the most affected.

To verify that these discrepancies are indeed due to the treatment of the extrema, we provide maps of $\alpha_{NO} + \alpha_{NO_2} + \alpha_{HNO_3} + \alpha_{TRCb}$ at the end of the simulation (Fig. 4). These maps show that the discrepancies are indeed stronger in the vicinity of the peak of $NO_x$ mixing ratios, and that the error patterns are extremely similar in all schemes.

✓ **Fig. 3 will be included in the revised version of this manuscript**.

With our best regards,

S. Mailler, R. Pennel, L. Menut, A. Cholakian

---

## Author Response (AR2)

**Author's response following acceptation as is for manuscript "An improved version of the Piecewise Parabolic Method advection scheme: description and performance assessment in a bidimensional testcase with stiff chemistry in toyCTM v1.0.1"**

Sylvain Mailler[1,2], Romain Pennel[1], Laurent Menut[1], and Arineh Cholakian[1]

[1] *LMD/IPSL, École Polytechnique, Institut Polytechnique de Paris, ENS, PSL Research University, Sorbonne Université, CNRS, Palaiseau France*
[2] *École des Ponts-ParisTech, Marne-la-Vallée, France*

Dear Editor,

We are very grateful to the Reviewers and the Editor for the proceeding of our manuscript. Since the editorial decision éfollowing our revised version of the manuscript is "publish as is", we have resubmitted the exact same version, just fixing an insignificant typo ("As *in* Fig. 6. . . " instead of "As *is* Fig. 6. . . " in the caption of Fig. 8).

With our best regards,

The authors.